# Repeat mediated excision of gene drive elements for restoring wild-type populations

**Pratima R. Chennuri**[1], **Josef Zapletal**[2], **Raquel D. Monfardini**[1], **Martial Loth Ndeffo-Mbah**[3,4], **Zach N. Adelman**[1], **Kevin M. Myles**[1] *

1 Department of Entomology and AgriLife Research, Texas A&M University, College Station, Texas, United States of America, 2 Department of Industrial and Systems Engineering, Texas A&M University, College Station, Texas, United States of America, 3 Department of Veterinary Integrative Biosciences, Texas A&M University, College Station, Texas, United States of America, 4 Department of Epidemiology and Biostatistics, Texas A&M University, College Station, Texas, United States of America

* mylesk@tamu.edu

**Data Availability Statement:** All files pertaining to the generation of equations, model execution, and plotting of charts can be found at the following link:

## Abstract

Here, we demonstrate that single strand annealing (SSA) can be co-opted for the precise autocatalytic excision of a drive element. We have termed this technology Repeat Mediated Excision of a Drive Element (ReMEDE). By engineering direct repeats flanking the drive allele and inducing a double-strand DNA break (DSB) at a second endonuclease target site within the allele, we increased the utilization of SSA repair. ReMEDE was incorporated into the mutagenic chain reaction (MCR) gene drive targeting the *yellow* gene of *Drosophila melanogaster*, successfully replacing drive alleles with wild-type alleles. Sequencing across the Cas9 target site confirmed transgene excision by SSA after pair-mated outcrosses with yReMEDE females, revealing ~4% inheritance of an engineered silent TcG marker sequence. However, phenotypically wild-type flies with alleles of indeterminate biogenesis also were observed, retaining the TGG sequence (~16%) or harboring a silent gGG mutation (~0.5%) at the PAM site. Additionally, ~14% of alleles in the F2 flies were intact or uncut paternally inherited alleles, indicating limited maternal deposition of Cas9 RNP. Although ReMEDE requires further research and development, the technology has some promising features as a gene drive mitigation strategy, notably its potential to restore wild-type populations without additional transgenic releases or large-scale environmental modifications.

## Author summary

CRISPR/Cas9-based autonomous homing gene drives have the potential to drastically reduce or eliminate vector-borne diseases, agricultural pests, and invasive species. However, their use raises significant regulatory, ethical, environmental, and sociopolitical concerns, particularly regarding unintended consequences. Responsible application of this powerful technology necessitates the development of control measures to halt or reverse any undesired outcomes. Several mitigation strategies have been proposed and, in some cases, demonstrated in laboratory settings, but their effectiveness in addressing concerns surrounding the use of gene drives in nature still remains uncertain. Therefore, ongoing

https://github.com/mln27/Python-CRISPR-1-1-SEM-gene-drive.

**Funding:** The National Institute for Allergy and Infectious Diseases and Defense Advanced Research Projects Agency supported this work through grants AI119081, AI148787 and HR0011-16-2-0036 to KMM and ZNA. The funders had no role in study design, data collection and analysis, decision to publish, or preparation of the manuscript.

**Competing interests:** I have read the journal's policy and the authors of this manuscript have the following competing interests: KMM and ZNA are inventors on US provisional patent application PCT/US2021/041951, submitted by Texas A&M University, which covers vector constructs that are pre-programmed to self-eliminate or self- remove at a predetermined time, and methods of making the same. PRC, JZ, RDM, MLNM declare no competing interests.

investigation and consideration of additional control strategies is prudent. This manuscript describes proof-of-concept studies for a novel gene drive mitigation strategy, which we have termed Repeat Mediated Excision of a Drive Element (ReMEDE). While further research and development is required prior to any possible application of this strategy, the ReMEDE concept offers a theoretical potential to restore transgene-free populations with wild-type phenotypes after a well-defined period of efficient gene drive.

## Introduction

Gene drives are currently being developed as tools for the control of sexually reproducing populations of organisms that negatively impact public health, agriculture, and conservation efforts [1,2]. The core technology underlying various gene drive-based control methods currently under development are transgenes that will be inherited at super-Mendelian rates, after the individuals that have been engineered to carry them mate with wild populations. Examples of organisms that are currently being considered for control with this technology include insects that adversely affect public health, e.g., mosquito vector species [3–5], agriculture, e.g., pest species [6,7]; and invasive rodents threatening the natural fauna and flora of island ecosystems, e.g., rat and mouse species [8,9]. Although the concept of large-scale genetic engineering projects to control or modify entire populations has existed for some time [10,11], the description and development of the type II CRISPR/Cas9 system for introducing targeted double-strand DNA breaks (DSBs) [12], with subsequent demonstration of the system's potential for inducing mutagenic chain reactions [3], has spurred a period of rapid technological development in this area. In theory, the CRISPR/Cas9 system is capable of cutting the DNA of any organism. Thus, many different gene drive configurations have been developed or theorized that are based on this programmable nuclease system originally from bacteria and archaea [2,13].

While none of these configurations have yet been tested in the field, the development of CRISPR/Cas9-based homing gene drives are proceeding rapidly [13]. In this configuration, the transgene encodes the Cas9 endonuclease and a guide RNA (gRNA). The synthetic gRNA will direct the Cas9 to a predetermined target site present in the organism's genome, resulting in a DSB at the target location. As eukaryotic organisms routinely repair DSBs, engineering the transgene with the sequences flanking the DSB (i.e., homology arms) enables a form of homologous recombination, termed homology directed repair (HDR), that is capable of restoring the integrity of the double helix structure at the break site using a nearby homologous donor sequence, usually a sister chromatid during the S and G2 phases of the cell cycle. While HDR is the most common of the various homology-based repair processes, it also requires the longest lengths of sequence homology between the donor and acceptor DNA. Homology directed repair of the DSB site, with the transgene serving as the donor, results in "gene conversion". If gene conversion occurs in the germline, the gene drive will be heritable, and in theory the process repeated indefinitely in subsequent generations, converting heterozygous progeny to homozygous progeny in the germline. This process ensures that the transgenic drive allele is inherited at rates far exceeding normal ratios of Mendelian inheritance. For this reason, homing gene drives have been referred to as being "threshold-independent", i.e., capable of being transmitted through a population following the introduction of an undefined but low number of individuals carrying the transgenic drive allele [14]. However, multiple studies that have mathematically modeled the spread of homing drives predict that in reality this type of drive may only be able to spread through a population when introduced

above a particular frequency, which is likely determined by a number of different factors, e.g., fitness cost of drive allele, naturally occurring resistant alleles, etc. [11,15–17].

Nevertheless, the specter of introducing, at previously unimaginable scales, potentially irreversible alterations to complex ecosystems raises a number of complicated ethical, legal, and social concerns [18]. Therefore, if the potential medical, agricultural and environmental benefits of gene drive technologies are to be fully realized, these concerns will need to be mitigated. This can be achieved through the development of technologies capable of limiting, both spatially and temporally, the spread of gene drives through a population [13,14]. While a number of mitigation strategies for minimizing the possibility of unanticipated harmful effects on humans, animals and the environment have been proposed [2,14,19–23], current state-of-the-art technologies for limiting CRISPR/Cas9-based homing gene drives are based on various designs of transgenic genetic elements [21] or split drive configurations [24–27]. A number of other systems have also been proposed, and in some cases demonstrated in laboratory studies, to limit the spread of CRISPR/Cas9-based homing gene drives [2,19,20,28,29]. Although demonstrating the efficacy of these systems is a significant advance, it is not yet clear that any of the current technologies will be able to adequately address concerns surrounding the use of gene drives in nature. While other types of drive mechanisms also exist [30], and may ultimately prove to raise fewer concerns than homing drives, no drive technology has yet been deemed entirely safe and tested in nature.

Previously, we proposed a novel approach to leverage a less common form of homology based repair, single-strand annealing (SSA), for the precise removal of gene drive alleles [31]. Theoretically, this technology could provide an autocatalytic method for removing the transgene during the gene drive process, ultimately leading to the restoration of the original wild-type population (Fig 1). However, our previous work in the area of gene drive removal by SSA was entirely computational. Here, we demonstrate proof-of-concept for this approach through the excision of an autonomous CRISPR/Cas9-based homing drive in the genetic model organism, *Drosophila melanogaster*. This was accomplished by co-opting the SSA repair mechanism, following induction of a DSB by a second endonuclease at a target site encoded within the drive allele (Fig 2). We increased the likelihood of SSA by engineering direct repeats at sites flanking the drive allele. A silent mutation (TGG > TcG) was engineered into the right homology arm of the construct in order to provide confirmation of transgene (i.e., the drive element) excision by SSA, but also functions as a silent engineered resistant allele in the critical protospacer adjacent motif (PAM) essential for Cas9-mediated cleavage (Fig 2). The altered PAM sequence prevents self-targeting that might lead to the formation of undesirable non-homologous end joining (NHEJ)-mediated resistant alleles or allelic conversion within the ReMEDE

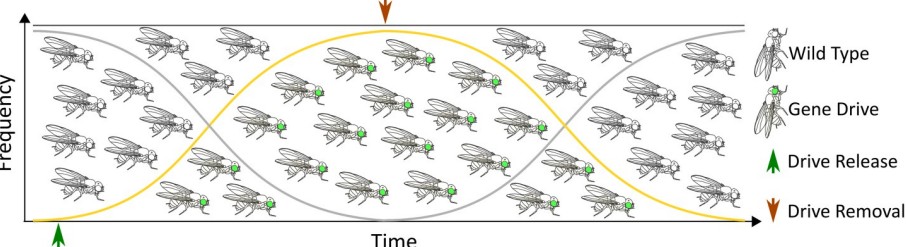

**Fig 1. Graphical summary.** Goldilocks scenario where a CRISPR/Cas9-based autonomous homing gene drive spreads through a target population over a defined period of time, after which the transgene is gradually removed through SSA restoring a wild-type phenotype.

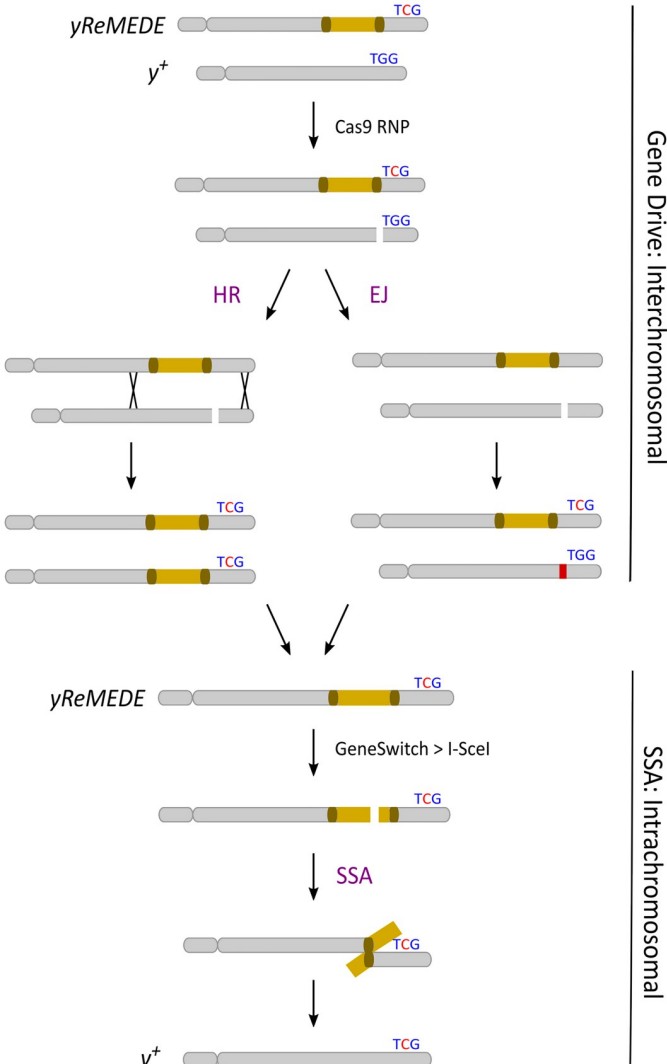

**Fig 2. Technology overview.** Schematic depicting homologous recombination (HR), end joining (EJ), and single strand annealing (SSA) in an autonomous homing gene drive containing the ReMEDE system.

allele. We have termed this technology Repeat Mediated Excision of a Drive Element (ReMEDE).

## Methods

### Ethics statement

All gene drive lines were handled in an ACL-2 facility in accordance with standard containment protocols. All work with genetically modified organisms was performed under protocols approved by the Institutional Biosafety Committee at Texas A&M University.

### Plasmid construction

Standard recombinant DNA techniques were used to clone all constructs. S1A (yDR and yRe-MET), S2A (yMCR) and S2B Fig (yReMEDE) outline plasmid construction. Complete maps and sequences are shown in S7 Fig.

## Microinjections

Plasmids were purified using the Macherey-Nagel NucleoBond Xtra Midi endotoxin-free kit (#740420) and sent to BestGene Inc. or GenetiVision for embryo microinjections. The plasmids p-yMCR.EGFP and p-yDR.EGFP were co-injected with transient sources of a single guide RNA targeting exon 2 of the *yellow* gene (pCFD3-dU6:3-y1-sgRNA) and Cas9 (pBS-Hsp70-Cas9, a gift from Melissa Harrison & Kate O'Connor-Giles & Jill Wildonger—Addgene plasmid # 46294) into a $w^{1118}$ stock (BDSC #3605). The plasmids p-ReMET.RFP and p-ReMEDE.RFP were co-injected with a transient source of φC31 into stocks carrying the construct p-yDR.EGFP. All transgenic stocks were confirmed through PCR and Sanger sequencing.

## Genomic DNA isolation and sequence analysis

Genomic DNA was prepared from individual flies according to protocols from Gloor et al., 1993 [23,32]. PCR reactions were assembled with GoTaq G2 Green Master Mix (#7823), purified with Zymo Research DNA Clean & Concentrator-5 kit (#4014), sequenced and analyzed with SnapGene and PolyPeakParser.

## Genetic crosses and phenotyping

Injected G0 transformants were crossed with $w^{1118}$ flies and G1 progeny screened for either EGFP or RFP positive individuals. These individuals were then outcrossed with $w^{1118}$ flies prior to establishing homozygous lines. The yMCR or yReMEDE males were crossed with wild-type ($w^{1118}$) females in order to generate F1 "master" females. The master females were then mated with wild-type males in single pair crosses for scoring F2 gene drive phenotypes.

## Population cage studies

Discrete small scale population studies were set up in standard fly rearing bottles (Genesee Scientific Cat. #32–130) with a design similar to those described previously [23,32]. Briefly, for each genotype an equal number of virgin females and males were seeded. All flies were removed following egg laying. The progeny from each subsequent generation were collected and then separated into two equal pools (n = 150) of randomly selected flies. Flies from one pool were screened and phenotyped, while the other pool was seeded into a new cage.

## Drosophila husbandry

The I-Site reporter line, $w^{1118}$; $P\{w^{+mW.hs} = w8,I\text{-}site\}7$ (BDSC# 6972), was sourced from the Bloomington Drosophila Stock Center. A nos-Gal4 line, $y^* w^*$; $P\{w^{+mC} = GAL4\text{-}nos.NGT\}40$ (Kyoto DGRC# 107–748), was obtained from the Kyoto Drosophila Stock Center. Stocks were maintained on standard oatmeal/cornmeal medium at 18˚C, but all experiments were performed at 25˚C, unless otherwise noted. Every 3rd generation, gene drive flies were scored and manually sorted to remove phenotypes consistent with resistant alleles before tipping into new vials and bottles.

## Results

### Repeat Mediated Excision of a Transgene (ReMET)

As an initial test of the hypothesis that the SSA repair pathway can be co-opted for the removal of gene drive alleles, we generated multiple transgenic lines containing ReMET reporter constructs flanked with SSA substrates (i.e., direct repeats) of various lengths (30–500 bp). Briefly,

the reporter constructs consist of an endonuclease, I-*Sce*I, under the control of a 5x Upstream Activation Sequence (*UASp.I-SceI*), the 18 bp I-*Sce*I target site (I-Site), and two fluorescent marker sequences, enhanced green fluorescent protein (EGFP) and red fluorescent protein (RFP), under the control of independent 3xP3 promoters. The yeast transcription activator protein, Gal4, was placed under the control of the germline-specific promoter *nanos* (*nos*) in a mifepristone (RU486)-inducible GeneSwitch (*nos*-GS) expression system (Figs 3A and S1A). Constructs were inserted into the *yellow (y)* gene of *D. melanogaster*. Expression of I-*Sce*I generates a DSB at the I-Site sequence, which facilitates SSA-based repair mediated by the presence of the direct repeats, leading to transgene excision. Successful ReMET in F2 progeny is indicated by the loss of yellow body color and both fluorescent marker sequences (Fig 3A). A silent mutation (TGG > TcG) was engineered into the right homology arm flanking the yReMET constructs so that a single nucleotide change would be left behind following SSA repair, providing evidence that the transgene had in fact been present prior to removal, which would otherwise be indistinguishable from wild-type sequences (Figs 3A and S1A). It should be noted that both the wild-type allele (TGG) and engineered allele (TcG) produce wild-type body pigmentation. Previous studies suggest that the length of direct repeats influences the efficiency of SSA in repairing DSBs [33–35]. To determine if a linear relationship exists between direct repeat length and SSA frequency in our system, we tested a range of substrate lengths. We generated individual yReMET lines featuring direct repeat sequences of 30, 250, or 500 base pairs (bp).

It is well known that the GeneSwitch system exhibits some leakiness [36]. Therefore, the stability of reporter lines containing direct repeats was initially investigated with ReMET constructs lacking the inducible *nos-GS*/GAL4 expression system (*in trans*). Importantly, we did not observe evidence of unintended recombination events after crossing these lines with wild-type flies (S1B Fig). However, pair-mated reciprocal crosses of F1 flies with a *nos-Gal4* driver line resulted in successful ReMET events (i.e, excision of the transgene mediated by SSA). The frequency of these events was consistent with a linear relationship between SSA and the direct repeat lengths flanking the transgene, with lower rates of excision observed for the 250 bp length and higher rates observed with the 500 bp length (Fig 3B). We did not observe SSA-mediated excision of the transgene with the 30 bp direct repeat length (Fig 3B), suggesting that this length may be at or below a minimum effective processing segment (MEPS) length [37]. Interestingly, SSA appeared to occur at a relatively higher mean rate in the male germline, with successful ReMET observed in ~15% of F2 progeny when the 250 bp direct repeats were present, and in ~35% of the F2 offspring when the 500 bp direct repeats were present. However, successful ReMET was observed in only ~6% of the F2 offspring of yReMET females for the 250 direct repeat length, and in ~15% of progeny when the 500 bp direct repeats were present (Fig 3B).

As expected, crossing siblings with the complete ReMET construct, which included *nos-GS*/GAL4 (*in cis*), in the absence of RU486 resulted in low levels of progeny (0.2 to 0.6%) exhibiting evidence of SSA (S1C Fig), due to leakiness of the GeneSwitch system [36]. However, a more surprising result was that significant increases in the frequencies of successful ReMET were not observed in the presence of RU486 (S1C Fig). ReMET was observed in ~1% of the F1 offspring with the 250 direct repeat length, and in ~0.5% of the progeny when the 500 bp direct repeats were present (Fig 3C). Similar to our earlier results, there was little evidence of SSA with the 30 bp direct repeat length (Fig 3C). It is possible that the relatively lower levels of SSA observed in these experiments are somehow related to the continual leaky expression of the endonuclease, I-*Sce*I, from the *nos-GS*/GAL4 system throughout development, rather than introduction through a parental cross, as was the case with the previous *in trans* configuration. Nevertheless, the presence of the silent molecular marker, TcG, engineered into the right

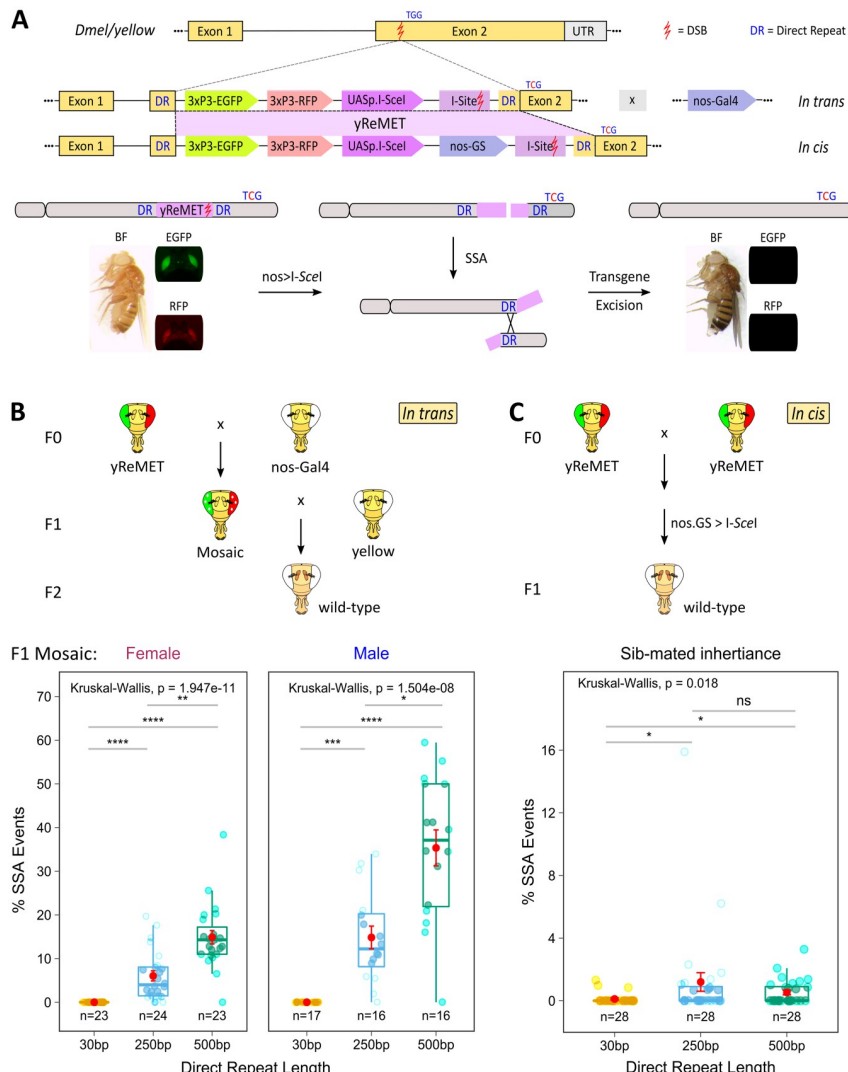

**Fig 3. Repeat Mediated Excision of a Transgene (ReMET). (A)** Schematic depicting ReMET constructs (in trans and in cis) and their removal by SSA. The yReMET construct (shaded in purple) contains EGFP and RFP under the control of independent 3xP3 promoters, and an endonuclease, I-SceI, under the control of UASp. The construct is flanked by a pair of direct repeats (DR) that are 30, 250, or 500 bp in length. Insertion of the ReMET construct into exon 2 of the *yellow* gene creates a null allele that produces yellow body pigmentation with both green and red eye fluorescence. I-SceI may be expressed by crossing with a nos-Gal4 driver line (in trans), or by expression from a RU486-inducible nos-GS (in cis). Expression of I-SceI generates a DSB at the recognition site (I-Site) located between the DRs, excising the transgene and one of the direct repeats through SSA. SSA-mediated removal of the direct repeats through SSA. SSA-mediated removal of the transgene restores wild-type body pigmentation (brown body) with simultaneous loss of eye-specific fluorescence. The engineered synonymous mutation TGG->TcG provides molecular confirmation of ReMET. **(B)** Schematic depicting in trans mating scheme and phenotypic identification of germline SSA events. Percentage of F2 progeny exhibiting SSA-mediated transgene excision by DR length, confirmed through wild-type body color and the presence of the engineered silent mutation (TcG). Each dot represents a separate pair-mated cross (mean and ± s.e.m. are shown in red), with the sex of the F1 yReMET fly shown above. **(C)** Schematic depicting in cis mating scheme and phenotypic identification of germline SSA events. Percentage of F1 progeny exhibiting SSA-mediated transgene excision by DR length, confirmed through wild-type body color and the presence of the engineered silent mutation (TcG). Each dot represents a separate pair-mated cross (mean and ± s.e.m. are shown in red). UTR = Untranslated Region; DSB = Double Strand Break; UASp = Upstream Activation Sequence with pTransposase promoter; EGFP = Enhanced Green Fluorescent Protein; RFP = Red Fluorescent Protein; SSA = Single Strand Annealing; p values = * < 0.5, ** < 0.01, *** < 0.001, and **** < 0.0001; ns = not significant.

homology arm provided sequence confirmation of ReMET in all flies exhibiting evidence of SSA (i.e., the loss of yellow body color as well as both fluorescent marker sequences). In order to confirm activation of the *nos-GS* by mifepristone, males containing the full ReMET construct were crossed with a previously published transgenic line [38], containing a *white* ($w^+$) reporter gene with an adjacent I-Site sequence, in the presence of increasing concentrations of RU486. As expected, cutting at the site adjacent to the $w^+$ reporter by I-*Sce*I, expressed from the ReMET cassette, deleted all or part of the $w^+$ gene resulting in eye-color mosaicism (S1D Fig). While I-*Sce*I was expressed at low levels from 5xUAS, even in the absence of treatment with mifepristone, increasing levels of mosaicism were observed in the eyes of F1 progeny depending on the concentration of RU486 present (S1D Fig). This confirms that the *nos-GS* system present in the full ReMET constructs is functioning properly. Thus, leaky expression of I-*Sce*I from the *nos-GS* system appears to induce SSA at a maximum level, explaining why additional activation with RU486 does not increase ReMET frequencies in the F1 progeny (S1C Fig). Nevertheless, these experiments demonstrate co-option of the SSA repair pathway for the precise removal of a transgene from *D. melanogaster*.

## Super-Mendelian inheritance of a CRISPR/Cas9-based autonomous homing gene drive

In order to extend our findings demonstrating removal of a transgene to an active gene drive (Fig 4A), we reconstructed the previously described CRISPR/Cas9-based yMCR gene drive [3,23]. This gene drive system is based on homology-dependent integration into the X-linked recessive *yellow* locus (S2A Fig). In gene drives, allelic conversion turns heterozygous recessive mutations into homozygous null mutations in the homogametic maternal germline (Fig 4B and 4C), leading to inheritance of the drive element at frequencies above 50% [3,23,39]. However, other genetic outcomes are also possible, including the formation of resistant alleles via repair of the Cas9 RNP-induced double-strand breaks by the more error-prone NHEJ pathway. Male flies with resistant alleles may show a wild-type or yellow phenotype, depending on the location of the mutation in the gene and how it affects the reading frame (Fig 4C). In the absence of Cas9, female progeny inheriting a null mutation typically would not display a yellow phenotype, as the presence of a single wild-type allele would be dominant (Fig 4C). However, following crosses of gene drive females with wild-type males, female progeny inheriting resistant alleles can exhibit a yellow phenotype or mosaic wild-type pigmentation on a yellow body. This is likely due to maternally deposited Cas9 RNP in the zygote, which repeatedly cleaves the paternal wild-type allele until NHEJ converts it into a mutant allele, leading to zygotic allelic conversion or somatic mosaicism (Fig 4C). Thus, while it is possible for in-frame mutations to result in wild-type phenotypes, out-of-frame alleles usually cause yellow pigmentation in female progeny (Fig 4C). This non-Mendelian exception, known as the "dominant maternal effect," has been documented in gene drive studies across species. [3,4,8,23,39,40].

We modified our MCR drive to include 3xP3-EGFP (MCR-EGFP) for tracking both functional and non-functional resistant alleles (S2A Fig). In 58 single pair-mated outcrosses between heterozygous F1 master females and wild-type males, the drive allele was transmitted to F2 progeny at an average rate of ~87% (Fig 4C). Approximately 11% of the offspring displayed a yellow mutant phenotype, while about 2% appeared wild-type (Fig 4C). These percentages are in line with previously documented yMCR inheritance patterns in *D. melanogaster* [3,23,39,40].

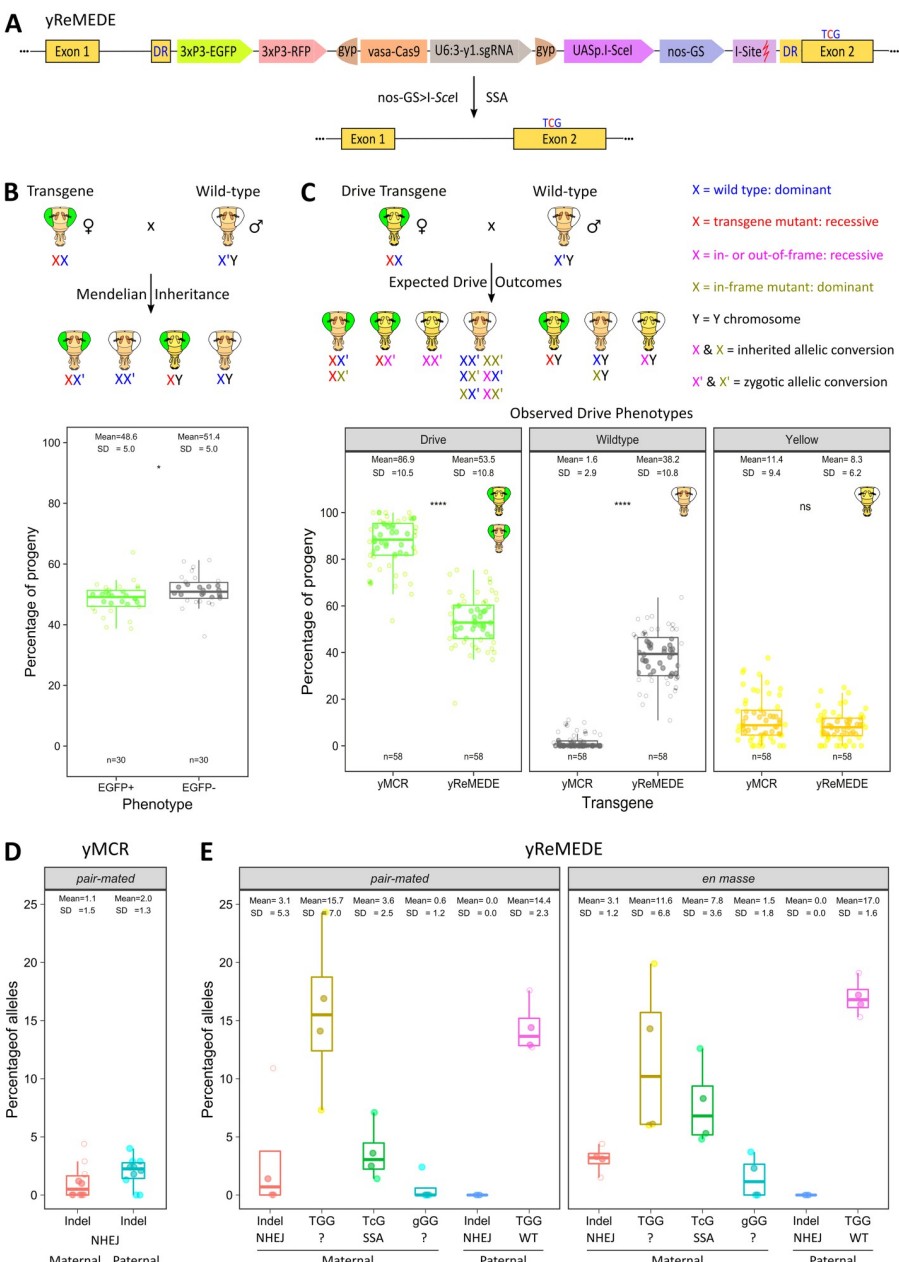

**Fig 4. Repeat Mediated Excision of a Drive Element (ReMEDE). (A)** Schematic depicting ReMEDE construct and its removal by SSA. The construct contains components of both the ReMET (in cis) and yMCR constructs, including those essential for gene drive, i.e., Cas9 (under the control of a vasa promoter) and an sgRNA targeting exon 2 of the wild-type *yellow* gene (under the control of a U6:3 promoter). The core gene drive components are flanked by a pair of gypsy insulator sequences (gyp). Expression of I-SceI from nos-GS generates a DSB at the I-Site located between the DR, resulting in excision of the transgene by SSA, which is confirmed by the presence of an engineered silent mutation, TcG. **(B)** Schematic depicting maternal inheritance of an X-linked transgene or gene drive lacking ReMEDE. Mendelian inheritance of the recessive X-linked *yellow* mutant allele (generated through transgene insertion) from a heterozygous mother produces wild-type body pigmentation in heterozygous female progeny, but a yellow phenotype in the hemizygous male progeny. **(C)** When the transgene is also a gene drive, multiple Mendelian outcomes, as well as non-Mendelian exceptions, are possible. When the heterozygous drive allele present in the F1 female creates a DSB in its wild-type counterpart, HR-mediated repair results in allelic conversion increasing transgene frequency and generating super-Mendelian patterns of inheritance (green eyes). However, DSBs repaired by the more erroneous NHEJ may generate either in- or out-of-frame recessive null alleles, or dominant functional in-frame alleles. Non-Mendelian exceptions may also occur due to the zygotic perdurance of maternally deposited Cas9 RNPs generating DSBs in the paternal wild-type alleles, resulting in either in-frame or out-of-frame repair, which

generates the additional possibilities illustrated. Theoretically, no wild-type X alleles should exist in the progeny of a cross involving a gene drive, which is why they are not shown. Maternal inheritance of the yMCR and yReMEDE transgenes. The percentage of progeny exhibiting drive, wild-type, or yellow phenotypes in replicate pair-mated crosses of heterozygous F1 females with wild-type males are shown. Each dot represents a separate pair-mated cross (**D**) Frequencies of out-of-frame indels in yMCR and yReMEDE progeny. Percentage of progeny by sex exhibiting a yellow body phenotype in pair-mated crosses involving either F1 yMCR or yReMEDE females and wild-type males. (**E**) Sequencing results for the *yellow* gene in wild-type progeny of pair-mated crosses involving F1 yMCR females and wild-type males, sorted by parental lineage. (**F**) Sequencing results for the *yellow* gene in wild-type progeny of pair-mated or en masse crosses involving F1 yReMEDE females and wild-type males, sorted by parental lineage. The sequence present at the PAM site is shown, as well as the DNA repair pathway presumed to be responsible for its generation. NHEJ = Non-Homologous End Joining; SSA = Single Strand Annealing; p values = ** < 0.01, *** < 0.001, and **** < 0.0001; ns = not significant (Wilcoxon signed-rank test).

## Repeat Mediated Excision of a Drive Element (ReMEDE)

The yMCR construct was next recombined into the yReMET (flanked by repeats of 250 bp) reporter line by standard methods to generate a yReMEDE line (Figs 4A and S2B). Based on results regarding direct repeat substrate length and SSA frequency (see above), all subsequent experiments employed a direct repeat length of 250 bp. F1 females were generated by crossing yReMEDE males to wild-type females. The performance of the yReMEDE system was then assessed in a parallel series of outcrosses (n = 58), where the heterozygous F1 yReMEDE females were pair-mated with wild-type males. The allelic inheritance patterns of the resulting progeny were then compared to those resulting from the outcrosses with the F1 yMCR females (Fig 4C). Both the genotypes and phenotypes of progeny that are expected to result from crosses involving F1 yReMEDE females are shown in S3 Fig. Successful ReMEDE is suggested by the loss of fluorescent marker genes and reversion to wild-type body pigmentation in the F2 progeny, which was observed (S3 Fig). The number of individuals with wild-type phenotypes increased significantly, comprising ~38% of the total yReMEDE progeny over all crosses (Fig 4C). This significant increase in wild-type progeny correlated with a significant decrease in the overall mean transmission rate of the drive allele (~54%; Figs 4C and S2C).

Pair-mated crosses with yMCR master females rarely produced wild-type phenotypes (Fig 4C and S1 Data). When they were observed, sequencing confirmed the phenotype to result from indels, generated either by NHEJ acting on a maternally inherited allele, or a dominant maternal effect acting on a paternally inherited allele (Fig 4D and S1 Data). Indel sequences resulting from these two processes occurred with similar frequency in the wild-type progeny of yReMEDE master females as well (Fig 4E and S1 Data). Thus, the presence of wild-type F2 yReMEDE flies in excess of this frequency suggested an approximate rate of SSA. However, confirmation of gene drive excision by SSA required sequencing over the Cas9 target site in all wild-type individuals for the silent mutation (TGG > TcG) engineered into the right homology arm of the construct. Sequencing of wild-type progeny confirmed the presence of the engineered silent mutation (TcG), which was not present in F2 yMCR flies, in ~4% of maternally inherited alleles in F2 yReMEDE flies (Fig 4E and S1 Data). Although we observed the TcG sequence in a relatively modest subset of maternally inherited alleles, those present in male progeny were necessarily contributed by the master females (Figs 4E and S3 and S1 Data). In males, X-linked inheritance is exclusively maternal, indicating excision of the gene drive by SSA (S2C and S3 Figs). Female progeny carrying alleles with TcG, which were also present, cannot be as definitively assigned to the maternal lineage, but the presence of the TcG sequence still suggests SSA. Unexpectedly, some flies displaying a wild-type phenotype also possessed maternally inherited alleles of unknown biogenesis, featuring either the wild-type TGG sequence (approximately 16%) or a silent gGG mutation (around 0.5%) at the PAM site (Figs 4E and S3 and S1 Data). Consistent with limited or no maternal deposition of Cas9 RNP,

paternally inherited wild-type alleles increased to ~14% in the F2 yReMEDE flies (Fig 4E and S1 Data), which was in contrast with the paternally contributed alleles in wild-type F2 yMCR flies that contained only indels (Fig 4D and S1 Data).

To investigate further the origin of alleles present in wild-type flies, we sequenced all of the progeny generated from a yReMEDE single pair-mated cross with two sets of primers (Pair-mated cross #4; S1 Data). The first primer set encompassed the PAM sequence, while the second set encompassed the entire transgene, with binding sites located outside of the construct's left and right homology arms (S2A and S2B Fig). The use of both primer sets permitted us to ascertain the zygosity of both drive (ReMEDE) and non-drive alleles (wild-type or containing indels), even in the homogametic female progeny. The results confirmed that in cross #4 only ~3% of the non-drive alleles in heterozygous females (i.e., wild-type phenotype) carried the engineered TcG marker, with a larger ~17% exhibiting the original wild-type TGG sequence (Pair-mated cross #4, S1 Data). In contrast, all ReMEDE drive alleles (see Fig 4C), present in either male or female progeny, were positive for the purposely engineered TcG variant of the PAM sequence (Pair-mated cross #4, S1 Data).

We initially considered the possibility that incorporating the ReMEDE components inadvertently deactivated Cas9, leading to uncut alleles. However, multiple observations are inconsistent with this hypothesis. First, in single pair-mated crosses (n = 30) involving flies with an EGFP transgene inserted at the same site in the yellow gene (yEGFP), over half of the F2 flies inherited the transgene at rates below 50%, with only one replicate exceeding 60% (Fig 4B and S1 Data). In contrast, inheritance of the yReMEDE allele ranged from 18% to 75%, with more than 60% transmission observed in 15 separate replicates (Fig 4C and S1 Data). Although the mean values were similar, yReMEDE allele inheritance was significantly higher than that of the yEGFP transgene (p = 0.01; S1 Data). Second, the proportion of progeny with resistant alleles (yellow body color) in yReMEDE F2 generations did not differ significantly from those in yMCR F2 flies (Fig 4C and S1 Data). If Cas9 were inactive, the transgenic construct would follow normal Mendelian inheritance, resulting only in EGFP-positive yellow males (Fig 4B). Third, yellow F1 females (often called "master" females) were produced by crossing yReMEDE males with wild-type females (Fig 4C), an outcome that required Cas9-mediated cutting of the maternal wild-type allele.

To further test this, we performed pair-mated crosses (n = 13) with hemizygous yMCR or yReMEDE males and wild-type females. In these crosses, all female offspring inherited the gene drive transgene and expressed fluorescent markers (S4A Fig). Since yellow body pigmentation is an X-linked recessive trait, it would appear in these females only if Cas9 from the paternal transgene was active. With no maternal Cas9 source, any loss of function in the maternally inherited yellow allele must have resulted from Cas9 expressed by the paternally inherited gene drive. If the yReMEDE drive produced lower levels of Cas9, to an extent that some alleles remain uncut, we would expect to observe this. However, this was not the case. Instead, as anticipated, 100% of female progeny from both yMCR and yReMEDE crosses exhibited yellow pigmentation, suggesting approximately equivalent Cas9 activity from both elements (S4A Fig). Interestingly, we did not observe any SSA outcomes in these crosses.

To assess SSA in the absence of drive-induced cleavage or leaky *in-cis* expression of I-*Sce*I, we performed crosses with a yReMEDE transgene containing mutations in the vasa promoter that render Cas9 non-functional (S4B Fig). Similarly, the nos-GS element driving I-*Sce*I expression was inactivated by replacing the UASp sequence with UASt. Unlike UASp in the active yReMEDE gene drive element, UASt is repressed in the female germline of *D. melanogaster* [41]. Homozygous females or hemizygous males for the transgene were outcrossed to either wild-type flies or a transgenic line expressing I-*Sce*I from the Hsp70 promoter. I-*Sce*I expression was induced by subjecting embryos to heat shock (38°C for 1 hour). Fluorescent

markers in yReMEDE were used to assess transgene inheritance. Mosaic F1 individuals were outcrossed to wild-type flies, and germline events were identified in F2 progeny by fluorescence. Crossing schemes for transgenic males and females are shown in S4C and S4D Fig. As expected, control crosses with wild-type flies resulted in Mendelian inheritance of the transgene in F2 progeny (S4C Fig). However, outcrosses with the I-*Sce*I line (experimental test) showed a significant reduction in F2 progeny with fluorescent markers, indicating SSA-mediated loss of the transgene (S4C Fig and S1 Data). The percentage of SSA events, categorized by sex, is shown in S4E Fig and S1 Data. Sequencing of randomly selected families from female experimental outcrosses confirmed that some alleles contained the engineered TcG sequence but were otherwise wild type, suggesting SSA events (S4F Fig and S1 Data). Similarly, sequencing from a male experimental outcross showed that all alleles in non-fluorescent progeny contained the TcG sequence, further suggesting SSA-mediated inheritance (S4F Fig and S1 Data).

Replicate *en masse* crosses involving multiple F1 yReMEDE females and wild-type males yielded an average transmission rate of ~53% for the drive allele, with ~42% of the progeny exhibiting a wild-type phenotype (S2D Fig), numbers very similar to those observed in pair-mated crosses (Fig 4C). Similar percentages were observed in a series of additional replicate crosses performed in the presence of RU486 (S2E Fig). Sequencing over the Cas9 target site in wild-type individuals for the silent mutation (TGG > TcG), engineered into the right homology arm of the construct, again provided confirmation of drive element excision by SSA (Fig 4E and S1 Data).

Periodic sequencing of ReMEDE fly stocks at the target site of the rare-cutting endonuclease I-*Sce*I (I-Site) over a period spanning ~1303 days, indicated that the cut site remained intact in sequenced flies for at least 691 days. At some point, between days 691 and 1303, the I-Site was disrupted by a 17 bp deletion (Figs 5A and S8). The stability of the I-Site is sequence is likely influenced by genomic location, and may exhibit lower stability at other genomic loci. Thus, strategies to minimize NHEJ at the I-Site cut site may need to be incorporated into future iterations of the ReMEDE technology. Nonetheless, these experiments demonstrate

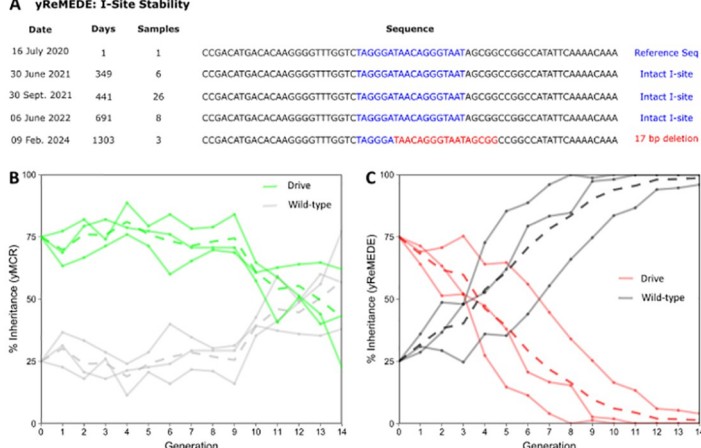

**Fig 5. I-Site stability and population cage trials with either yMCR or yReMEDE gene drives. (A)** Representative sequences over time at the I-*Sce*I (I-Site) target site in the yReMEDE line (actual sequences are provided as a SAM file in supplementary information in S8 Fig). The samples column indicates the number of individuals sequenced on a particular date, with exception of the most recent samples, which represent 3 different mosquito pools, each consisting of 15 males and 15 females sequenced by nanopore. **(B)** Graphs show the prevalence of drive and wild type phenotypes (exhibited by both male and female flies) over time in cage trials involving either yMCR (green lines) or **(C)** yReMEDE (red lines). Solid lines represent single replicate cages, while dotted lines represent mean values.

proof-of-principle for successful co-option of the highly conserved SSA repair pathway for precise autocatalytic excision of an active autonomous homing gene drive.

## ReMEDE in population cage studies

In order to evaluate the ReMEDE system over multiple generations, we conducted cage trials with either yReMEDE or yMCR drive flies. The cage trials were performed in accordance with those described in previously published studies [23,32]. We evenly divided six hundred male and female virgins into three replicate cages (200/cage). Within each cage, 25% of the flies were wild type and 75% homozygous for either yMCR or yReMEDE. At each generation (n), approximately 150 flies/cage were randomly selected for scoring (i.e., fluorescent markers and body color), and another randomly selected 150 flies/cage seeded into the subsequent generation (n + 1) of cages. Additional details are provided in S1 Text. The frequency of flies with unmodified yMCR alleles (EGFP+) decreased only modestly, still representing a significant portion of the population after 14 generations (Fig 5B). In contrast, the frequency of flies exhibiting yReMEDE drive phenotypes (i.e., EGFP+, RFP+, and yellow body color) decreased gradually, and were completely eliminated in 10–14 generations (Fig 5C). Reciprocally, the frequency of flies exhibiting a wild-type phenotype increased gradually, representing 100% of the population after 10–14 generations (Fig 5C). These experiments suggest that all gene drive elements carrying ReMEDE could eventually be converted to alleles producing a wild-type phenotype given enough generations.

## ReMEDE-mediated conversion of a threshold-independent gene drive

A major drawback of drives targeting the *yellow* gene is that loss-of-function adversely affects male mating success [23,42–46]. Alleles containing indels, generated either through NHEJ or dominant maternal effect, occurred with similar frequency with both the yMCR and yReMEDE drives (Fig 4C). However, wild-type flies that arise through the formation of resistant alleles have an enormous courtship advantage, which results in elimination of the drive allele in population cage studies. This process plays out even more rapidly in the presence of ReMEDE, which is of course designed to generate silent resistant alleles producing a wild-type phenotype (Fig 5B and 5C). Therefore, drives targeting the *yellow* gene are not ideal for demonstrating the ReMEDE technology in the context of population studies; although, this target site does have some other advantages. Thus, we sought to generate mathematical models to predict the dynamics of a low or no-threshold drive containing ReMEDE over multiple generations. In order to do this, we first attempted to determine the mating cost associated with *yellow* mutant flies, containing an insertion at the same site targeted by the drive allele; methods detailed in S1 Text.

   *D. melanogaster* engages in a series of courtship rituals (e.g., tapping, chasing, genital licking, singing, etc.) prior to successful copulation. A previous study has shown that the lower mating success of yellow males in comparison to wild-type males does not result from any observable difference in the amount of time spent courting, but rather from delayed initiation of copulation [45]. However, these experiments were conducted with Canton-S wild-type flies. In our population studies, it was necessary to use the $w^{1118}$ strain as wild type, in order to score eye-specific fluorescence. The *white* (*w*) gene has also been associated with copulation success, and $w^{1118}$ flies display reduced courtship activity [47]. Thus, we calculated courtship indices (the proportion of time that a male was engaged in courtship activities divided by the total duration of observation) for wild-type ($w^{1118}$) and mutant ($w^{1118}$;*yellow*) flies. Although wild-type males displayed reduced courting activities with wild-type females, the courtship of mutant *yellow* females was even more markedly reduced (S6A Fig). Male *yellow* mutants had

no success mating with wild-type females in these trials; thus, a courtship index was not recorded for these crosses (S6A Fig). However, the courtship index of mutant *yellow* males was approximately 0.5, when calculated from crosses with mutant *yellow* females. Given the results of the single pair courtship assays, mating costs associated with mutant *yellow* males were instead calculated from competitive assays, where a wild-type virgin female was provided with both a wild-type male and mutant *yellow* male. In the 30 crosses performed for these assays, a single *yellow* male, exhibiting a courtship index of ~0.5, successfully copulated (S6A Fig). Based on the results of these experiments, we estimated the mating cost associated with mutant *yellow* males to be 29/30 or 0.97 (S6A Fig).

We have previously generated mathematical models for SSA-based excision of transgenes [31,48]. These initial parameters were used to simulate allele frequencies for both autosomal and X-linked inheritance patterns, and then modified to include empirical values derived from crosses involving yMCR ($\delta$ = NHEJ and mating cost of 0.97 for *yellow* males) and yReMEDE ($\alpha$ = SSA, $\gamma$ = SSA resistance, and $\epsilon$ = engineered allele persistence) flies. Methods are detailed in S1 Text and visualized in S6C Fig. The model was then used to run simulations of various scenarios. In conjunction with these simulations, we conducted multi-generation population cage trials to assess the impact of mating costs and resistant alleles associated with the *yellow* gene target site of the yMCR gene drive. Replicate cages were seeded with flies, such that yMCR females or males comprised 10% of the population, with the other 90% being virgin wild-type males or females; a level that might be consistent with the introduction of a threshold independent drive element. The scoring of each generation (n) and seeding of subsequent generations (n + 1) was performed as described above. In each of the cages, individuals exhibiting fluorescence and yellow body phenotypes were rapidly lost from the population, regardless of sex (S6B Fig). Simulations conducted with similar parameters matched the observed performance of the drive element (g) in population cage studies (S6D Fig). In an attempt to remove the mating costs associated with the *yellow* gene target site, additional cages were seeded with flies in which yMCR females or males again comprised 10% of the population, but with the other 90% being males or females homozygous for the null *yellow* allele (i.e., a mutant population). The frequency of the EGFP+ cassette (i.e., the yMCR drive allele) increased in the population, ultimately achieving 15–60% penetrance when introduced by female flies, and 30–80% penetrance when introduced by the male flies (S6B Fig). Simulations of male releases derived with similar parameters roughly approximated the trends observed in the population cage trials (Fig 6A).

Additional mathematical modeling removing all mating costs associated the yReMEDE drive, and with SSA (female $\alpha$ = 0.1, male $\alpha$ = 0.01) occurring at ~10%, where all events are marked (TcG) by the engineered resistant allele (v; $\epsilon$ = 1), results in a maximum ~50% drive allele frequency after about 15 generations, which are then progressively replaced by drive-resistant TcG-marked wild-type alleles (v) (Fig 6B). However, several studies have analyzed and overcome the challenge of resistant alleles, either by targeting essential genes, using multiple guide RNAs, recoding target genes, or by incorporating a combination of these strategies to ensure that only homozygous drive or wild-type allele combinations remain viable [5,24,40,49]. In lieu of these empirical developments, we simulated yReMEDE population dynamics in the absence of both mating costs and resistant alleles ($\delta$ = 0), with three different SSA frequencies. If approximately 30% of maternally inherited alleles are formerly drive alleles that were converted to wild-type alleles by SSA, of which 25% retain the engineered TcG resistant allele (v; $\epsilon$ = 0.25), w alleles (native wild type) and v alleles (drive-resistant wild type) persist in the population, while the drive alleles (g) are progressively eliminated (Fig 6C). Alternatively, when SSA occurs at ~10%, with all events retaining the TcG marker ($\epsilon$ = 1), allelic conversions of wild-type alleles (w) by drive alleles achieve a maximum frequency of ~60%

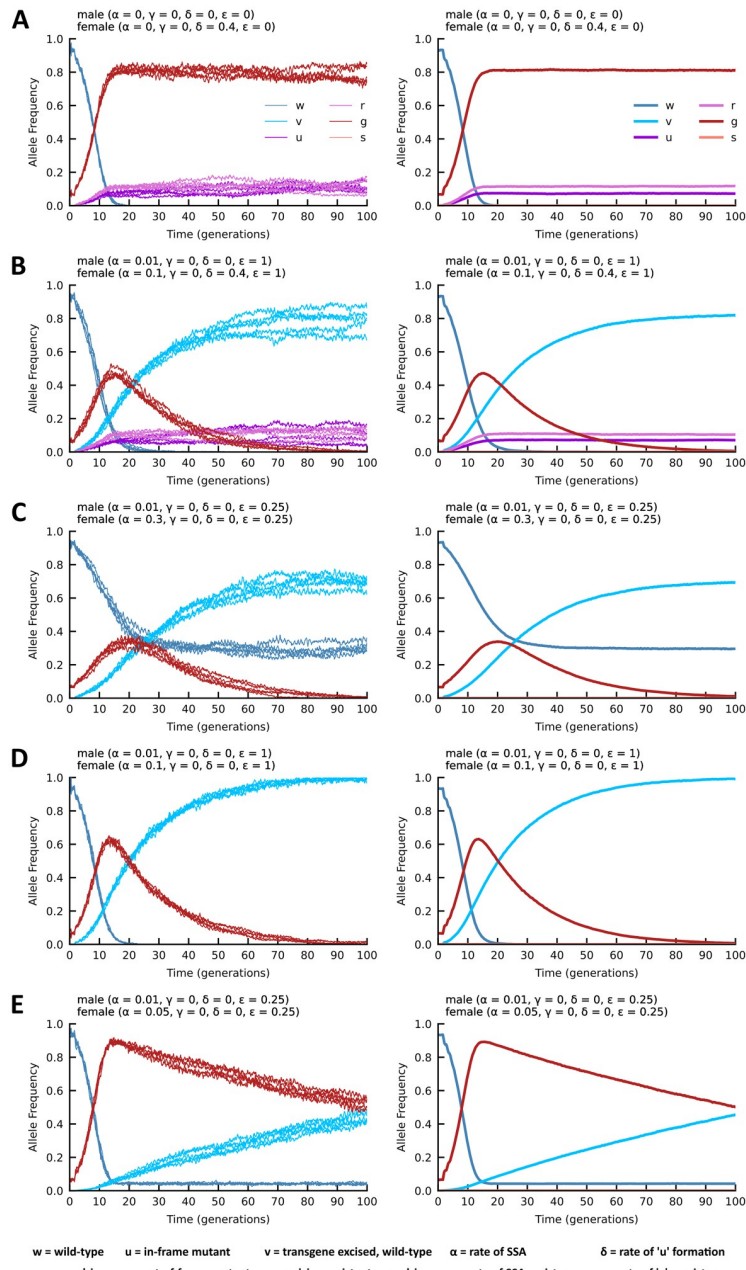

w = wild-type          u = in-frame mutant          v = transgene excised, wild-type          α = rate of SSA          δ = rate of 'u' formation

g = gene drive          r = out-of-frame mutant          s = excision resistant gene drive          γ = rate of SSA-resistance          ε = rate of 'v' persistence

**Fig 6. Stochastic modeling of ReMEDE technology performance in a population.** All models were run with the X-linked inheritance module that outputs allele frequencies in two plots. One plot shows five simulations randomly selected from 100 independently run simulations **(Left),** while the other plot shows the mean of all simulations **(Right).** All simulations were run in a low threshold release scenario (10% of flies containing a drive allele) and in the absence of mating costs. **(A)** Simulation of yMCR gene drive, generating resistant alleles at frequencies comparable to those observed experimentally, but in the absence of reduced courtship activity and mating success. **(B)** Simulation of yReMEDE where SSA converts approximately 10% of the drive alleles to wild type. The generation of resistant alleles and other parameters occur at frequencies comparable to those shown in A. **(C)** Simulation of a gene drive containing ReMEDE that generates no resistant alleles, but with SSA converting approximately 30% of the drive alleles to wild type; a quarter of which retain the engineered resistant allele in the PAM site ($\varepsilon = 0.25$). **(D)** Simulation of a gene drive containing ReMEDE that generates no resistant alleles, but with SSA converting approximately 10% of the drive alleles to wild type, all of which retain the engineered resistant allele in the PAM site ($\varepsilon = 1$) **(E)** Simulation of a gene drive containing ReMEDE that generates no resistant alleles, but with SSA converting a modest ~5% of the drive alleles to wild type, a quarter of which retain the engineered resistant allele in the PAM site ($\varepsilon = 0.25$).

in < 15 generations, but are then subsequently entirely replaced by drive-resistant alleles (v) via SSA (Fig 6D). When SSA occurs at the minimal rate of ~5%, with only 25% of these retaining the TcG marker ($\varepsilon$ = 0.25), the frequency of the drive allele (g) quickly reaches ~90% in < 15 generations, but is subsequently replaced by engineered drive-resistant alleles (v) after several more generations (Fig 6E). To account for potential NHEJ-driven disruption of the I-Site (allele 's'), which would render SSA inoperable, we simulated a low rate (0.1%) of I-Site disruption ($\gamma$ = 0.0005x2 = 0.001). The rate chosen reflects the relative stability of the I-Site sequence observed in empirical studies (Fig 5A). The simulation suggests that drive alleles would be replaced with the engineered allele 'v' even in the presence of low-level accumulation of SSA resistant alleles (S6E Fig). However, it must be noted that the success of the strategy would most certainly be compromised if the rate of I-Site disruption became too high. Nonetheless, stochastic modeling (informed with empirical data) indicates that 5–10% of SSA events, where only one-fourth of the events remain drive-resistant (v), would initially permit efficient propagation of a low-threshold drive, with eventual replacement by the engineered drive-resistant alleles.

## Discussion

Early successes mitigating the negative effects of resistant alleles on drive efficiency [5,40,50–52] suggest that the development of technologies for limiting the spread of drives through a population may be necessary, if such tools are ever to be successfully tested and deployed in the field. Indeed, several technological solutions have recently been proposed for homing drives, with the design and demonstration of various braking systems and confinable gene drive configurations representing a particularly positive development in this regard [2,19–22,28,29]. Nonetheless, it remains unclear if any of the currently proposed technologies will ultimately prove to be an adequate solution. First, many of these strategies are premised on the introduction of additional independent transgenic genetic elements. These elements share a number of similarities with gene drives themselves, and as a result their efficacy may be influenced by the same parameters as those affecting the homing drives they are designed to stop (i.e, gene conversion efficiency, fitness costs, timing, etc.), and indeed this has been observed in several laboratory trials and simulations [11,15–17,23]. Second, the introduction of additional transgenic elements may complicate risk assessment, as it might not be possible to extrapolate with a high enough degree of confidence how an autonomous drive might function in a particular ecological environment based on field testing in a split configuration, or in the presence of an independent genetic braking element. Third, no strategy satisfactorily addresses the critical requirement for removal of transgenes from natural populations, should that become necessary. Rather, the removal of multiple independent transgenes at the conclusion of a field trial, in any sort of acceptable time frame, would likely require inundation through the sustained release of wild-type individuals. We, therefore, sought to develop a solution where the excision of the transgene would be autocatalytic over the period of gene drive, eventually restoring the wild-type population. By design, drive activity is meant to be gradually reduced as drive alleles are converted to wild-type alleles. We have previously proposed co-opting the SSA repair mechanism for this purpose [31,48].

Here, we demonstrate proof-of-concept using a self-eliminating gene drive [3] targeting the *yellow* gene of *D. melanogaster*. The system, which we termed ReMEDE, significantly increased the proportion of progeny displaying a wild-type phenotype. Crosses between master yMCR females and wild type males rarely resulted in wild type phenotypes (Fig 4C and S1 Data), which is consistent with previous findings with the yMCR drive, where in-frame functional mutations were also uncommon [23,32,39]. Sequencing did not reveal any alleles with a wild

type TGG PAM sequence in these flies (Fig 4D and S1 Data), suggesting that non-drive alleles are repeatedly cleaved by Cas9, leading to either allelic conversion or resistant allele formation. The frequencies of functional (wild-type phenotype) and non-functional (yellow phenotype) indels present in the F2 progeny of yReMEDE and yMCR master females were very similar (Fig 4C, 4D and 4E and S1 Data), suggesting similar levels of Cas9 activity in both drives. Rather, the primary difference between the F2 generations was the presence of wild-type phenotypes containing alleles with either TGG, gGG, or TcG sequences at the PAM site in the yReMEDE progeny, which were not observed in the yMCR progeny (Figs 4D and 4E and S1 Data). Sequencing showed only the intentionally engineered TcG variant in the PAM sequence of intact ReMEDE alleles (Pair-mated cross #4, S1 Data).

Unlike the yMCR progeny where the wild-type PAM site in all paternal alleles was converted into an indel, paternally inherited alleles in yReMEDE offspring remained intact, (Fig 4E and S1 Data). This suggests that maternal deposition of Cas9 RNP was inhibited in yReMEDE flies (Figs 4D, 4E and S2C and S1 Data). One possibility is that the ReMEDE mechanism, removed the gene drive, preventing perdurance of the Cas9 RNP. Alternatively, the inclusion of gypsy insulators in the ReMEDE construct might have somehow decreased maternal deposition of Cas9, which may have led to a lower rate of zygotic allelic conversion. Insulators, such as those derived from the gypsy retrovirus, have been shown to block the effects of adjacent enhancers, silencers, and heterochromatin—collectively referred to as position effects—thus significantly increasing transgene expression levels in comparison with non-insulated loci. In particular, the gypsy retrotransposon insulator has been demonstrated to boost gene expression to levels several-fold greater than those observed in loci without insulators [53]. Thus, the idea that gypsy elements reduce Cas9 expression in this system seems unlikely but cannot be entirely ruled out. Another explanation might be that the presence of the gypsy insulators leads to more precise Cas9 expression, which if proven would be significant.

In these experiments, "master females" were generated by mating a drive male with a wild-type female to effectively eliminate any maternal Cas9 deposition. Although we cannot entirely exclude the possibility that the TcG sequence in the drive allele served as a template for repairing the wild-type allele, results from the inactive yReMEDE transgene suggest this is unlikely. In male outcrosses involving the inactive transgene, non-fluorescent female progeny were produced, all of which carried alleles with the TcG sequence (S4D Fig). These observations suggest that repair likely occurred through the combined action of I-*Sce*I, direct repeats, and SSA. Overall, the results from the yReMEDE experiments indicate that both drive conversion and SSA contribute to the observed outcomes.

The absence of SSA in pair-mated crosses with hemizygous yReMEDE males suggests that gene drive removal occurs only in the female germline when I-*Sce*I is expressed *in cis* (S4A Fig). This absence of SSA may be attributed to transcriptional silencing of I-*Sce*I in the germline of heterogametic males. The ReMEDE construct was designed to promote inter-chromosomal allelic conversion in homozygous individuals (females), whereas SSA was expected to operate intrachromosomally in both homozygous females and hemizygous males. However, in crosses between yReMEDE males and wild-type females, no female wild-type offspring displaying the engineered TcG allele were observed (S4A Fig). All male progeny were wild type, as the X chromosome is inherited from the female parent (S4A Fig). Transcriptional silencing of transgene expression on the X chromosome relative to autosomes is a well-described phenomenon [54]. SSA can occur in both the germline and the zygote. Silencing occurs in both premeiotic and meiotic cells and is governed by mechanisms independent of meiotic sex chromosome inactivation (MSCI) [54,55]. SSA occurs during the maternal-to-zygotic transition (MZT), which begins around mitotic cycle 8 and becomes widespread by cycle 14 [56,57]. If SSA occurred during the MZT, it would likely affect only a subset of nuclei, potentially

resulting in mosaic phenotypes, none of which were observed in this case. Therefore, the absence of SSA events on the paternal X chromosome in the female offspring of yReMEDE males may be due to X-chromosome inactivation in the male germline and the limitation of zygotic events to a small number of cells (S4A Fig). Indeed, SSA was only observed in the progeny of males, carrying the inactive yReMEDE transgene, when I-*Sce*I was provided *in trans* (S4E Fig). However, the sequence and length of the transgene to be excised may also influence the frequency of SSA.

Transcriptional silencing of I-*Sce*I in the germline of heterogametic males may also explain differences in the efficiency of transgene excision observed in the previously described experiments with the yReMET-*cis* and yReMET-*trans* constructs (Fig 3B and 3C). However, explaining the differences in apparent SSA rates between the gene drive experiments and those involving the yReMET transgene, where I-*Sce*I was also expressed *in cis*, is more challenging; these differences may be linked to the gene drive process itself. Interestingly, in some crosses involving flies with the yReMET-cis transgene, SSA rates were significantly higher than those observed with the yReMEDE gene drive, based on measurements using the engineered marker allele (Fig 3C). This indicates that SSA rates varied considerably among these crosses. Therefore, one possible explanation is that in the presence of the gene drive, SSA becomes more consistent, leading to the higher average rate observed (Fig 4E).

Sequencing wild-type offspring from yReMEDE master females in pair-mated crosses revealed two unexpected yellow allele genotypes (TGG or gGG). It remains uncertain whether these genotypes arose from SSA repair or if the yReMEDE gene drive has a lower cutting rate compared to the original yMCR. The latter seems more likely, possibly due to gypsy elements reducing Cas9 expression or the ReMEDE mechanism excising the drive before sufficient Cas9 is expressed to cut all wild-type alleles. However, previous studies suggest that gypsy elements should not significantly reduce Cas9 expression, given their well-established insulator function in *D. melanogaster* [53]. Moreover, although we lack direct evidence showing that yReMEDE expresses Cas9 at the same levels as yMCR, some of our results are inconsistent with the idea that lower expression is responsible for the unexpected genotypes. For instance, crosses of F1 yMCR females with wild-type males produced indels in 1–2% of inherited alleles (Fig 4D), and similar frequencies (~3%) were observed in F1 yReMEDE females (Fig 4E). These comparable rates are difficult to reconcile with a scenario where yReMEDE expresses Cas9 at a significantly lower level than yMCR. Similarly, crosses of yReMEDE males with wild-type females showed that 100% of the female offspring exhibited yellow pigmentation, as observed in male crosses with yMCR (S4A Fig), further suggesting comparable Cas9 activity between yReMEDE and yMCR. Therefore, it seems more likely that the ReMEDE mechanism excises the drive before sufficient Cas9 can be expressed to cut all wild-type alleles. However, other possibilities exist. The dual nuclease system in yReMEDE may cause genome damage during the copying of the drive cassette to a wild-type target, potentially reducing the fraction of chromosomes carrying the drive construct. This could occur if some of the damaged alleles present in females were homozygous lethal (or hemizygous lethal in males).

Another alternative hypothesis is that, following ReMEDE, the engineered silent TcG marker is sometimes converted to TGG or gGG in the F1 heterozygous maternal germline through synthesis-dependent strand annealing (SDSA) and mismatch repair (MMR; see S5 Fig for the model). Although alleles containing gGG were exceedingly rare, their occurrence is difficult to explain with the previously outlined hypotheses. There are two generally accepted models for the repair of DSBs by HR, double strand break repair (DSBR) and synthesis-dependent strand annealing (SDSA). Both models begin with the resection of the 5' termini flanking the DSB. This generates long single-stranded 3' tails, one of which invades the homologous duplex template sequence resulting in the formation of a D-loop. After this step, the two

models diverge wherein the DSBR model is characterized by formation of double Holliday junctions (dHJs), the resolution of which may result in either crossover or non-crossover products [58]. SDSA is mediated by new synthesis, disassociation of the recently extended strand, and annealing with the other 3' tail in the broken chromosome, exclusively generating non-crossover products (S5 Fig). While the DSBR model appears to be preferred in meiotic cells, evidence increasingly suggests that the SDSA model is the major pathway of DSB repair in mitotic cells [59]. In some cases, MMR machinery appears to be capable of recognizing and repairing mismatches present in the transient heteroduplex DNA (hetDNA) intermediates formed during the repair process [60]. However, the newly synthesized strands, produced in both the DSBR and SDSA models, may also form hetDNA that can be detected and repaired through MMR. In either case, loss of heterozygosity would be possible through nonreciprocal transfer of information from the template strand [61]. In the ReMEDE system, DSBs generated by I-*Sce*I occur between engineered direct repeats increasing the likelihood that they will be repaired by SSA, whereas those generated by Cas9 do not occur between direct repeats making it more likely that HDR will be chosen. Occurrences of wild-type PAM sequences (TGG) in the yReMEDE lines, but not in yReMET lines that were devoid of Cas9, might suggest that it is the cleavage mediated by Cas9 that is necessary for SDSA, and subsequently MMR, resulting in reversion of the engineered marker TcG to TGG. Sequencing of progeny from single pair-mated crosses revealed that all of the ReMEDE drive alleles present retained the engineered silent mutation, suggesting that these reversions would have to happen following excision of the ReMEDE construct (S1 Data).

Our previous work on the removal of gene drives using single-strand annealing (SSA) was entirely conceptual, making it impossible to predict how the technology would perform within an actual gene drive in a biological organism. However, one concern that we considered a distinct possibility was that SSA might be too efficient at removing the transgene, to the point that gene drive might not occur. In order to address this possibility, we placed the second nuclease (I-*Sce*I) under the control of an inducible gene switch system. However, this system is leaky, and we continue to see expression of I-*Sce*I (S1D Fig). To our knowledge, all inducible systems exhibit some level of leakiness. In any case, these concerns ultimately proved to be unfounded, as SSA occurred at relatively low levels, even when higher levels of the nuclease were produced (S1C Fig). These results suggest that there may be no need for an inducible system in future iterations of the technology, which in any case would have been difficult to implement in a field setting. In practice, SSA re-constitutes wild-type alleles at relatively low levels, which is desirable as the wild-type population is only gradually restored, permitting a period of gene drive. Thus, it may be possible to achieve a "goldilocks" scenario, where gene drive is balanced against SSA in order to permit the drive to achieve a sufficient level of spread through a population prior to removal. Alternatively, the technology may be better suited for application as a braking system (i.e., *in-trans* configuration).

In multi-generational population cage studies, the ReMEDE drive element was completely eliminated, and a phenotypically wild-type population restored in 10 to 14 generations (Fig 5C). A number of factors related to the performance of the yMCR gene drive (i.e., mating defects, resistant allele formation, etc.) prevented us from conducting population-level studies under conditions that would mimic the release of a low-threshold gene drive (S6 Fig). However, simulations with experimental parameters obtained from this study suggest a similar dynamic would occur with low threshold drives. Simulations of yReMEDE population dynamics, in the absence of both mating costs and resistant allele formation, determined that only 5% of SSA-mediated recombination events, even with only a fraction of these retaining the engineered PAM site (if SDSA and MMR are occurring), would be sufficient to permit the initial propagation of a low threshold drive, followed by its eventual replacement with the engineered

alleles (Fig 6E). These frequencies are not far off of those observed in single pair mating experiments, suggesting that the ReMEDE technology may be capable of halting more robust gene drives, eventually reverting the population back to a wild-type phenotype. Given the limitations of the yMCR gene drive, future work will focus on employing the ReMEDE system in the context of more robust gene drives.

The yReMEDE construct proved to be relatively stable. Sequencing did not reveal any instability in the construct, other than at the I-Site sequence (Fig 5A). However, the genomic target site location (i.e., the *yellow* gene) may play a role in the relative stability of the construct. The *yellow* gene is present on the X-chromosome and males are hemizygous for X. Also, male Drosophila do not undergo meiotic recombination and also exhibit transcriptional suppression of the X chromosome. Additionally, the F1 females generated from male parents are generally heterozygous. These might be mitigating factors influencing the stability of the construct, but testing the ReMEDE system in the context of other gene drives will provide more information. Nonetheless, the present study constitutes an effective proof-of-principle for the ReMEDE concept as a viable technology that merits further consideration in controlling autonomous homing gene drives. The ReMEDE system has a number of theoretically desirable features as a technological safeguard for gene drives, including the potential to restore a phenotypically wild-type population, and address the issue of transgene persistence. The ReMEDE concept should be considered along with other strategies currently under investigation for controlling, neutralizing, halting, deleting, etc., gene drives. Particularly, as early modeling efforts suggest that threshold-independent drives, similar to those that have already been developed in the malaria vector *Anopheles gambiae*, may be difficult to control with braking systems [62]. While other strategies have also been demonstrated in laboratory settings, it is too soon to know if any of these will be successful in field settings [2,19,21,28]. Further, it is entirely plausible that no mitigation strategy will prove ideal for every possible scenario involving gene drives. Thus, prudence would appear to dictate that multiple gene drive mitigation strategies are pursued further at this juncture.

## Supporting information

**S1 Data. Numerical data underlying graphs and summary statistics provided in spreadsheet form.**
(XLSX)

**S1 Fig. Generation of yReMET lines, transgene excision, and characterization of RU486 inducible promoter.** (**A**) Schematic depicting the two-step process for the generation of yReMET (*in cis*) transgenic lines. In the first step, a donor plasmid, p-yDR.EGFP, with DRs of varying lengths (30, 250 or 500 bp), identical in sequence to the 3' end of the 5' homology arm, were inserted into the *D. melanogaster yellow* gene at a Cas9 target site. The donor construct contained 3xP3-EGFP, which served as a marker of transgenesis, a recognition site for the I-*Sce*I endonuclease (I-Site), a loxP site, and an attP landing site. The resulting yDR transgenic fly lines were also engineered to contain the silent PAM mutation TGG->TcG. In the second step, a donor plasmid, p-ReMET.RFP, containing an attB site was inserted into the yDR lines through φC31-mediated recombination. The p-ReMET.RFP donor contained 3xP3-RFP, a loxP site, as well as the nos-GS and UASp.I-*Sce*I components. In another configuration, used for the *in trans* experiments, only the UASp.I-*Sce*I is included and not the nos-GS. Integration of the p-ReMET.RFP plasmid into the yDR lines generated the various yReMET lines containing DRs of different lengths. (**B**) Percentage of F2 progeny exhibiting SSA-mediated transgene excision by DR length and sex, confirmed through the presence of wild-type body pigmentation and the engineered silent mutation (TcG). Gold dots represent individual pair-mated

crosses of F1 yReMET (*in trans* configuration) flies with wild-type flies. Blue dots represent individual pair-mated crosses of F1 yReMET (*in trans* configuration) flies with transgenic nos-Gal4 flies. P-value **** < 0.0001; ns = not significant. (**C**) Percentage of F1 progeny exhibiting SSA-mediated transgene excision by DR length and RU486 concentration, confirmed through the presence of wild type body pigmentation and the engineered silent mutation (TcG). Each dot/triangle represents a separate pair-mated cross (mean and ± s.e.m. are shown in red). (**D**) Images of adult eye pigmentation in F1 progeny resulting from pair-mated crosses of yReMET (*in cis* configuration) flies with flies containing a *white* ($w^+$) reporter gene with an adjacent I-Site sequence. Images are representative of the average level of mosaicism observed for the given concentration of RU486.
(TIF)

**S2 Fig. Generation and characterization of yReMEDE line.** (**A**) Schematic depicting insertion of p-yMCR.EGFP into the *yellow* gene of *D. melanogaster*. The plasmid p-yMCR.EGFP contains Cas9 (under the control of a vasa promoter), an sgRNA targeting exon 2 of the wild-type *yellow* gene (under the control of a U6:3 promoter), and EGFP (under the control of the eye-specific 3xP3 promoter). The primer annealing sites 43 and 44 are located outside the homology arms in order to permit genotyping of target alleles in flies exhibiting a wild-type phenotype. (**B**) Schematic depicting the generation of the yReMEDE line through φC31-mediated recombination of p-ReMEDE.RFP into the yDR line. The plasmid p-ReMEDE.RFP contains the yMCR drive element flanked by a pair of gypsy insulators (gyp), nos-GS > UASp-I-*Sce*I, a loxP site, and 3xP3-RFP. Primer annealing sites 208 and 16 permit genotyping of the PAM site in the drive allele. (**C**) Maternal and paternal inheritance of drive elements. Percentage of progeny exhibiting scored phenotypes (i.e., body pigmentation and fluorescent eye color) as indicated by cartoon fly heads. Each dot represents a separate pair-mated cross. (**D**) Percentage of progeny exhibiting wild-type phenotypes in replicate *en masse* crosses of yReMEDE flies. Each dot represents a separate *en masse* cross (mean and ± s.e.m. are shown in red). (**E**) Percentage of progeny exhibiting wild-type phenotypes in replicate *en masse* crosses of yReMEDE flies in the presence of RU486 at the concentrations specified. LHA = Left Homology Arm; RHA = Right Homology Arm; p values = ** < 0.01, *** < 0.001, and **** < 0.0001; ns = not significant (Wilcoxon signed-rank test).
(TIF)

**S3 Fig. Matrilineal inheritance of wild-type alleles in progeny of females carrying yReMEDE.** Mating scheme for generation of F1 yReMEDE females and their F2 progeny. Images of fly heads indicate scored phenotypes of eye fluorescence (white = no marker, green = EGFP, red = RFP) and body color (brown = + or +Δ, yellow = Drive or -Δ). Possible allelic combinations are shown below head images (ReMEDE = Drive, wild type = +, in-frame indel = +Δ, out-of-frame indel = -Δ). Male flies are indicated by the presence of the Y chromosome. Blue boxes enclose genotypes and phenotypes that confirm SSA-mediated transgene excision, regardless of the PAM sequence.
(TIF)

**S4 Fig. Patrilineal inheritance of yReMEDE and removal of inactive yReMEDE via Single-Strand Annealing (SSA).** (**A**) Schematic illustrating the expected genetic outcomes and associated phenotypes resulting from patrilineal inheritance of yReMEDE. (**B**) Diagram depicting the inactive yReMEDE construct and its removal through SSA after expression of the I-*Sce*I endonuclease (provided in trans) under the control of the Hsp70 promoter. (**C**) Schematic of the *in trans* mating scheme and phenotypic identification of germline SSA events. Transgenic F0 males were crossed with either wild-type (control) or *Hsp70 > I-SceI* (test) females.

Embryos were subjected to heat shock at 38˚C for 1 hour to induce *Hsp70*-driven expression of *I-SceI*. Germline SSA events occurring in $F_1$ individuals were identified by scoring $F_2$ offspring for fluorescent markers. Statistical significance was determined using the exact binomial test (*P*-values: ns, not significant; ****, $P < 0.0001$). (**D**) Schematic of the reciprocal cross where transgenic F0 females were crossed with either wild-type (control) or *Hsp70 > I-SceI* (test) males. (**E**) Percentage of SSA events observed in maternal or paternal germlines, estimated by comparing the number of fluorescent and non-fluorescent progeny. *P*-value: **** = $P < 0.0001$ (Wilcoxon signed-rank test). (**F**) Representative sequencing data confirming the presence of the engineered TcG-resistant allele in all non-fluorescent progeny from a randomly selected family from either female or male test crosses.
(TIF)

**S5 Fig. Model for reversion of the engineered TcG PAM sequence to wild-type.** The SDSA model appears to be the primary DSB repair mechanism in mitotic cells and is shown for simplicity. Production of Cas9 RNP from the gene drive (donor) allele (shown in green) generates a DSB in the wild-type (recipient) allele (shown in red). Resection of the 5' DNA ends produces 3' single-stranded tails. Strand invasion of the donor drive allele by one of the recipient allele's 3' tail results in base pairing of complementary strands generating a tract of heteroduplex DNA (hetDNA). This process displaces the originally duplexed strand forming a displacement (D)-loop. Elongation of the invading 3' end increases the size of the D-loop, eventually producing a sequence that extends beyond the DSB. Dismantling of the D-loop frees the newly synthesized sequence to anneal with the other 3' tail of the recipient allele, generating a new tract of hetDNA with the opposite strand, i.e., SDSA. Mismatches present in the hetDNA are repaired by MMR. Excision followed by resynthesis and ligation repairs the mis-paired bases, restoring the original wild-type sequence of the recipient and eliminating heterozygosity at the locus. SDSA = Synthesis Mediated Strand Annealing, DSB = Double Strand Break, RNP = RiboNucleoProtein.
(TIF)

**S6 Fig. Performance of the yMCR gene drive in a low threshold release scenario.** (**A**) Courtship indices calculated for *yellow* mutant (mt) and *w^1118* (wt) flies in the crossing schemes shown. P-values = ** < 0.01, and **** < 0.0001 (Wilcoxon signed-rank test). (**B**) Cage trials with the yMCR gene drive in either a *w^1118* (wt) or *yellow* mutant (mt) population. F1 females or males carrying the yMCR allele initially made up 10% of the total population. Graphs show the prevalence of the drive phenotype (green line) in both male and female progeny over time. Each line represents a separate replicate cage trial. (**C**) Theoretical model of allelic conversion dynamics occurring within the F1 female germline during oogenesis. In gene drive, a DSB occurs independently at each susceptible target allele (w) with a probability of *q*, after which repair occurs by HR with a probability of *p*, or by NHEJ with a probability of *1-p*. HR results in allelic conversion of w to g, while NHEJ produces in-frame alleles (u) with a probability of δ, or out-of-frame alleles (r) with a probability of 1-δ. In the presence of ReMEDE, a second DSB occurs and is repaired by SSA with a probability of α, or by NHEJ with a probability of γ. SSA retains the engineered silent PAM mutation (TcG) with a probability of ε, or TcG reverts back to TGG through MMR with a probability of 1-ε. (**D**) Stochastic modeling of the yMCR gene drive in a wild-type population. All models were run with the X-linked inheritance module that outputs allele frequencies in two plots. One plot shows five simulations randomly selected from 100 independently run simulations (Left), while the other plot shows the mean of all simulations (Right). Simulations of the yMCR gene drive were run in a low threshold release scenario (10% of flies containing a drive allele), and generating resistant alleles at frequencies (δ = 0.4) comparable to those observed experimentally. Similarly, mating cost parameters were set

at levels consistent with empirically determined values for yellow male (0.97) and female (0.31) flies in a wild-type population. (**E**) Simulation of a gene drive containing ReMEDE that generates no resistant alleles, but with SSA converting a modest ~5% of the drive alleles to wild type, a quarter of which retain the engineered resistant allele in the PAM site ($\varepsilon = 0.25$). Additionally, there is a low rate (0.1%) of I-Site disruption ($\gamma = 0.0005 \times 2 = 0.001$) from NHEJ. (TIF)

**S7 Fig. Sequences and maps of all plasmids used in this study.**
(DOCX)

**S8 Fig. Sequences of *I-SceI* target site in yReMEDE line over time.** SAM files of Sanger or nanopore sequences from time periods spanning from the inception of yReMEDE line through approximately 3.5 years.
(SAM)

**S1 Text. Supplementary methods.**
(DOCX)

## Acknowledgments

We thank Dr. Omar Akbari at the University of California San Diego for reviewing the manuscript and providing helpful comments. The graphic cartoon art of Drosophila Head Front (Somers, J. (2020). Zenodo. https://doi.org/10.5281/zenodo.3926059) and Drosophila Side (Somers, J. (2020). Zenodo. https://doi.org/10.5281/zenodo.3925951) used in the figures were downloaded and modified under Creative Commons Attribution 4.0 International. The graphical summary was modified from Chennuri *et.al* (2022) Front. Bioeng. Biotechnol. 10:897231. doi: 10.3389/fbioe.2022.897231 under the Creative Commons Attribution License (CC BY).

## Author Contributions

**Conceptualization:** Pratima R. Chennuri, Josef Zapletal, Zach N. Adelman, Kevin M. Myles.

**Data curation:** Pratima R. Chennuri, Kevin M. Myles.

**Formal analysis:** Pratima R. Chennuri, Josef Zapletal, Martial Loth Ndeffo-Mbah, Kevin M. Myles.

**Funding acquisition:** Zach N. Adelman, Kevin M. Myles.

**Investigation:** Pratima R. Chennuri, Josef Zapletal, Raquel D. Monfardini, Martial Loth Ndeffo-Mbah, Kevin M. Myles.

**Methodology:** Pratima R. Chennuri, Josef Zapletal, Martial Loth Ndeffo-Mbah, Kevin M. Myles.

**Project administration:** Pratima R. Chennuri, Kevin M. Myles.

**Resources:** Kevin M. Myles.

**Supervision:** Kevin M. Myles.

**Validation:** Pratima R. Chennuri, Kevin M. Myles.

**Visualization:** Pratima R. Chennuri, Kevin M. Myles.

**Writing – original draft:** Pratima R. Chennuri, Josef Zapletal, Martial Loth Ndeffo-Mbah, Zach N. Adelman, Kevin M. Myles.

**Writing – review & editing:** Pratima R. Chennuri, Kevin M. Myles.

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
