## [Decision Letter · Decision Letter 0]

11 Jan 2024

Dear Dr Myles,

Thank you very much for submitting your Research Article entitled 'Repeat mediated excision of gene drive elements for restoring wild-type populations' to PLOS Genetics.

The manuscript was fully evaluated at the editorial level and by independent peer reviewers. The reviewers appreciated the attention to an important problem, but raised some substantial concerns about the current manuscript. Based on the reviews, we will not be able to accept this version of the manuscript, but we would be willing to review a much-revised version. We cannot, of course, promise publication at that time.

If you decide to revise the manuscript for further consideration at PLOS Genetics, please aim to resubmit within the next 60 days, unless it will take extra time to address the concerns of the reviewers, in which case we would appreciate an expected resubmission date by email to plosgenetics@plos.org.

We are sorry that we cannot be more positive about your manuscript at this stage. Please do not hesitate to contact us if you have any concerns or questions.

Yours sincerely,

Jackson Champer

Academic Editor

PLOS Genetics

Gregory Copenhaver

Editor-in-Chief

PLOS Genetics

We have surprisingly received a large number of reviews for this manuscript despite the proximity of the winter holiday to the submission. I’d like to thank the reviewers for their rapid response during this time of year.

The topic and method of this study was interesting to myself and the reviewers. However, there are a large number of essential points detailed by the reviewers that I agree with and should be taken into account in a major revision before the paper can be published. Acceptance of the paper will ultimately be dependent on fully addressing all review comments, which in most cases can be by substantial changes and additions to the text.

While several points will require toning down efficiency claims in the manuscript, lower than ideal efficiency isn’t necessarily an obstacle to publication here. The fastest path to publication probably involves following most/all suggestions from the reviewers in these cases (you can always work to improve the efficiency in future studies).

There is some repetition between the reviewers (though each one also raises unique points), so in these cases, later responses in your “response to reviewers” document can be abbreviated, referring to previous responses.

Sincerely,

Jackson Champer

Reviewer's Responses to Questions

**Comments to the Authors:**

Reviewer #1: In this study, the authors proposed an innovative design (ReMEDE) to restrict the spread of CRISPR/Cas9-based homing gene drives by employing the SSA repair pathway to remove the drive element. The manuscript is well constructed, the experiments are reasonably designed, and the results are clearly described. Overall, this is an interesting and well-done study. It fits the publication scope of PLOS Genetics; thus, I recommend it to be published if the authors can address the below concerns.

#1: The authors claimed that the drive allele could be replaced by wildtype allele through SSA. In fact, there was a silent mutation TcG, instead of the real wildtype TGG, left in the genome after SSA. Though leading to a wildtype phenotype, the replaced allele could not be called a wildtype allele. The authors should emphasize this difference and redefine the allele in the manuscript.

#2: the end of page 2, the statement may not be quite accurate. “threshold-independent” gene drive should be able to spread in the population even with low initial release frequency. If they need a specific introduction frequency to spread, they are “threshold-dependent”, which could be achieved by adjusting modeling parameters as mentioned in the cited references. But in most cases or parameter range, homing drives are threshold-independent.

#3: Though SSA is a preferred repair pathway in these ReMEDE lines due to the flanking repeats, NHEJ is also a possibility in the repair of the I-SceI induced DNA break. This will generate resistance at the I-SceI target site, which may further spread in the population at a high frequency and impair ReMEDE. I would like to see sequencing evidence of any NHEJ present or absent in this system.

#4: As shown in the manuscript, a single I-SceI cut was not enough to always remove the drive allele, even with decent length flanking repeats. Hence, incorporating two I-SceI sites near both ends may help with removing such a large fragment, if they can be cut simultaneously. The other solutions could be using rescue element to eliminate resistant alleles with high fitness cost, or applying CRISPR coupled with multiplexed gRNAs instead of I-SceI.

#5: page 7, “Further, the overall mean inheritance of the transgene was significantly lower than that of the ReMEDE construct (p value = 0.01; Fig. 2B and C)”. ReMEDE should be corrected as yMCR.

#6: near the end of page 7, a low level of maternal deposition was detected by seeing GFP- yellow flies, but it still could not rule out the possibility that Cas9 was much less active (but not inactive at all) in the ReMEDE germline compared to yMCR, which caused low germline cut rate in the first place. It is possible that gypsy elements decreased Cas9 expression.

#7: in Figure 3, presenting allele frequencies (or wild-type versus ReMEDE SSA alleles if possible) at the end of the cage experiment would enhance the clarity of the results.

#8: some references may be cited imprecisely in the discussion session. Hammond et al 2017 focused on characterizing resistant alleles, while Unckless et al 2017 presented a modeling paper without demonstrating successful mitigation of resistance alleles.

#9: based on Figure 2, the germline cut rate in yReMEDE seemed quite low, though about 15% maternal deposition was detected. In Figure 2E, TcG was only detectable in maternal lineage rather than paternal lineage, which indicates that TcG might be derived from the early embryonic cut induced by Cas9/deposition from grandmother. Thus, it is possible that, in the drive mother, double strand break used TcG in the drive as a template and copied it to the original wildtype allele after Cas9/gRNA cut.

#10: the inheritance rate of yReMEDE was ~50% in crosses. With such low inheritance rate, the drive will not be able to spread in the population, just as shown in Figure 3 cage experiments. But in the modeling (Figure 4), why did all the drive increase in the first few generations? This result looks a bit confusing to me.

Reviewer #2: This paper is a proof of principle study for removing transgene drive alleles and restoring a phenotypical wildtype population using an inducible SSA repair pathway. Although this study proves that drive conversion and excursion of drive elements can happen simultaneously and produce more wildtype progeny, its medium drive efficiency and low SSA rate might limit its application. The modelling simulation suggests that drive alleles can be replaced by the engineered resistant allele over generations even with a low SSA rate when ignoring fitness cost and any other form of resistant allele.

Some specific comments on the manuscript:

1. In Figure S3, WT progeny is indicated as the result of the SSA event. However, as the authors discussed on page 13, these can also be uncut alleles, especially considering all ReMEDE drive alleles contain that TcG maker, and how WT alleles are repeatedly cut by Cas9 to form indel resistant alleles in yMCR crosses. Crossing the drive with an external source of Cas9 might provide some insight into whether or not the Cas9 is adequately expressed. This issue also applies to using the increase in WT progeny in yReMEDE crosses as a proxy for the SSA rate as stated on page 8 and later in the discussion on page 12. The percentage of WT progeny increases from ~2% to ~38% in Figure 2C, but nearly 29% of them are WT without the engineered TCG maker. It’s important to give a clear statement of the rate of SSA events under various scenarios and include it in the abstract so that readers can have a general impression at just a glance.

2. Figure 4: In figure 4B, authors included functional, non-functional and engineered drive resistant alleles. However resistant alleles can also form from NHEJ repair at the I-SceI cutting site. This type of resistant allele will prevent DSB thus SSA, allowing the drive allele to maintain or even spread, if the drive conversion rate allows, in the population. Do authors see any evidence of I-SceI resistance and at what frequency?

3. I recommend rearranging figures so that all information necessary for understanding this paper is presented in main figures, and supplementary figures don’t present the same information repeatedly. Now readers have to switch between main figures and supplementary figures multiple times while reading just one paragraph, e.g. the last paragraph on page 7. Also in Figure 2C, those yellow/brown fly legends only apply to yMCR crosses, but the graph also contains yReMEDE result and can be confusing.

Minor comments:

1. In-frame mutation does not guarantee to be functional alleles, as indicated in multiple locations.

2. What’s the estimated number of progeny for each generation in population cage?

3. On page 9, near the top, it’s Fig. S2E not S2D

Reviewer #3: Chennuri et al. test a gene drive reversal system in Drosophila that they proposed previously in a theoretical paper (Zapletal et al., 2021). The concept of this system, which is based on a large and expanding literature in yeast, mammalian cells, mosquitoes, and flies is based on a DNA repair pathway referred to as Single Strand Annealing (SSA). This repair pathway can trigger recombination between two directly repeated sequences in the genome if one induces a double strand DNA break (DSB) between the repeats, which results in a clean deletion of sequences between them. The authors had proposed and now test whether a system might be used to cleanly eliminate a Cas9-based gene drive system that was flanked by direct sequence repeats by engineering a gene drive to carry a second nuclease (I-SceI) that cuts at a target site also included in the gene drive element lying between the two directly repeated sequences. They refer to this system as a "Repeat Mediated Excision of a Drive Element" (ReMEDE).

The authors begin by inserting a series of fluorescently marked backbone constructs lacking CRISPR components but carrying an I-SceI target site flanked with differing lengths of direct repeats into the the Drosophila yellow locus at a site used for a previously well-studied gene drive system. Insertion of these cassettes interrupts the coding region of the yellow locus resulting in yellow mutant phenotype. They generate two forms of the repeat-flanked constructs, one carrying both the I-SceI nuclease under control of a drug inducible promoter as well as the I-SceI target site (cis version), and another lacking I-SceI but carrying the nuclease target site (trans version). The authors also introduce a silent sequence polymorphism just outside of the 3' direct repeat that can be used to distinguish cis deletion events from a reference WT allele. The trans version can then be combined with a GAL4-UAS regulated form of I-SceI to induce a DSB at the target site within the test construct. SSA events can be detected by elimination of two fluorescent marker genes as well as restoration of a WT pigmentation phenotype (and presence of the linked sequence polymorphism by PCR and sequence analysis). They compare the efficiencies of these two systems and make several observations. First, using the trans version, the authors observe frequent I-SceI induced deletion of the gene cassette which increases for repeats of 500 bp (~35%) compared to those of 250 bp (~15%). Shorter 30 bp repeats yielded few SSA events. Second, they note that the frequency of such expected cis-generated events is substantially higher (> 2 fold) in F2 offspring derived from gene cassettes carried by male versus female F1 parents carrying the gene cassette and I-SceI. Third, they observe much low frequencies of SSA for the cis version of the construct and these low frequencies do not increase even upon drug induction of the I-Sce1 transgene. The authors attribute this lack of inducibility to leaky drug-independent expression of the I-Sce1 transgene, although why these levels are so much lower than for the trans arrangement remains unexplained.

The authors then use Phi-C31 mediated recombination to insert core gene drive components into a cassette carrying the cis acting components of the I-SceI SSA system (including the flanking direct repeats) at the same site in the yellow locus. The resulting composite element carryies Cas9, a guide RNA for copying the drive element, I-SceI, and the I-SceI target site (note that the authors do not generate an important control element carrying both nucleases in the absence of the direct repeats - see critique). They compare the drive performance of the core drive (marked with eGFP) with the derived ReMEDE element and find that including the hypothesized leaky I-Sce1 cassette decreases the observed biased drive from ~87% for the core element to nearly Mendelian levels for the ReMEDE element. They also observe a much greater fraction of WT alleles for the ReMEDE element compared to the core drive which is more pronounced in female than male progeny by nearly a factor of 2. The authors also perform sequence analysis of progeny with a WT phenotype to determine whether they carry the expected cis-linked sequence polymorphism expected to be associated with SSA induced events. Curiously they find a mix of different linked allelic marker sequences that suggest something more complicated than simple SSA must be taking place. The authors propose a mix of copying of the gene element and SSA may be occurring.

Finally, the authors conduct a multi-generation population experiment in which they seed cages with 75% drive (core or ReMEDE) and 25% WT. They observe a gradual reduction in the frequency of the core drive and a corresponding increase of WT alleles over 14 generations. This result is not expected since the drive is expected either copy itself or generate NHEJ mutations that should have a mutant yellow phenotype as observed in their pair mating experiments (see critique below). In contrast, the ReMEDE element disappears steadily and completely over the course of 14 generations. The authors conclude that the ReMEDE system can promote restoration of a WT phenotype under these conditions, although this experiment does not probe a key question of whether this system might be employed in tandem with successful drive of an element followed by its elimination with ReMEDE.

Major critique points:

The concept behind this study of employing a repeat flanked drive system with SSA to restore a population to a WT phenotype is interesting and merits experimental testing as was the worthy goal of this study. Unfortunately, the data presented in the current manuscript to not provide sufficient information to test this idea in an informative fashion. There are several aspects of this study would require greater analysis to arrive at a proper evaluation of this methodology as summarized in the points below.

1) It is not clear why the cis acting version of the I-SceI bearing element is so much less efficient than the trans version or than observed in a variety of other prior studies in which SSA is used for a similar purpose to delete a gene cassette (see minor points below). Since the authors go on to use this inefficient cis configuration in their ReMEDE construct understanding why is a significant issue and bears on all subsequent results in the study. One way the authors could solve this problem is to include a different conditional system for expressing I-SceI such as a nos>GAL4 driven I-SceI (the same used for their active trans version) plus a temperature sensitive GAL80 inhibitor. Another approach would be to use a CRE-LOX type of system (which works very well in Drosophila) to flip out a conditional transcriptional termination cassette.

2) It is not clear that the results the authors report for the core drive versus the ReMEDE element are due to activity of the I-SceI system. This concern arises from several reasons including:

i) The issue raised above regarding the low SSA efficiency of the cis-acting I-SceI cassette.

ii) Most critically, the absence of proper control drive constructs that lack the I-SceI cassette (or essential parts of it) in the presence of the CRISPR components (as indicated in Fig. S2, the authors added the CRISPR components to the repeat flanked I-SceI element using Phi-C31 recombineering but did not generate a repeat flanked core drive element lacking the I-SceI components). This is a serious concern since one could wonder whether all the observed results with ReMEDE construct are actually due to Cas9-mediated cleavage and recombination induced with either cis or trans repeat sequences (indeed the result seemingly indicated by their polymorphism sequence analysis) rather than by I-SceI induced SSA as the authors infer.

iii) Another good construct control would include Cas9, guide RNA, I-SceI nuclease plus I-SceI target site and no direct repeats. This construct would assess the potential role of dual nucleases potentially creating damaged target alleles as may be the case given that WT progeny are nearly twice as abundant in females that could carry such alleles in a heterozygous condition than in males where such alleles would be lethal.

3) Why do WT alleles persist in the gene drive cage experiments using the core drive element which the authors show either copies itself (87%) or generates NHEJ mutations (8-11%) in single generation crosses? Are they using a cut resistant WT allele of yellow? The results and figure legend do not specify whether this might be the case. If so, is the same WT cut resistant target allele being used for the ReMEDE construct. If this is the case, the experiment shown in Fig. 3 is not very informative since there would be no Cas9-mediated target cleavage. Perhaps I am missing something?

4) The authors should consider and experimentally evaluate the possibility mentioned above (point 2, section iii) that a combined action of Cas9 plus I-SceI cleavage could lead to damaged chromosome targets in which a partial copy event mediated by Cas9 was then cleaved by I-SceI leading to DNA repair failure and potential chromosome-scale damage. One way to address this important question would be to use visible or dominant markers that are closely linked to the drive element to distinguish donor from receiver chromosomes.

5) A very significant limitation of the current study is that the authors do not test a ReMEDE system in the context that they are imagined to be useful namely in which a gene drive is allowed first to spread through a population (with the ReMEDE system off) and then have that cassette removed cleanly by switching the ReMEDE system on. Several suboptimal features of their system preclude such a key test including the absence of conditional activation of the ReMEDE system. This is really is an essential experiment and without it, one cannot judge the potential of the ReMEDE concept. As suggested in Point 1, the problem of developing a robust conditional system for activating I-SceI activity is one that needs to be solved in its own right for proper evaluation of ReMEDE as a viable system.

In summary, in my view, the study does not adequately resolve key mechanisms underlying the results they observe nor allow one to evaluate the potential of the ReMEDE type of drive reversal system. I therefore do not recommend publishing this manuscript in its current form. If the authors provide additional experimental evidence to resolve the issues raised above, a revised manuscript including these new results may merit re-evaluation.

Minor Points:

1) The genetic crossing scheme and color code for differing alleles in Fig. 2c are difficult to understand since the WT alleles in the parents are indicated by a blue maternal X or paternal X', but no such symbols are shown in the progeny. If all such alleles are assumed to being mutated then that should be explained and how that relates exactly to the different gray shaded X or X' alleles.

2) Several prior references demonstrating SSA in insects should be cited by the authors:

1. Grigoraki L, Cowlishaw R, Nolan T, Donnelly M, Lycett G, Ranson H. CRISPR/Cas9 modified An. gambiae carrying kdr mutation L1014F functionally validate its contribution in insecticide resistance and combined effect with metabolic enzymes. PLoS Genet. 2021;17(7):e1009556. Epub 2021/07/07. doi: 10.1371/journal.pgen.1009556. PubMed PMID: 34228718; PubMed Central PMCID: PMCPMC8284791.

2. Li X, Bai Y, Cheng X, Kalds PGT, Sun B, Wu Y, et al. Efficient SSA-mediated precise genome editing using CRISPR/Cas9. FEBS J. 2018;285(18):3362-75. Epub 2018/08/08. doi: 10.1111/febs.14626. PubMed PMID: 30085411.

3. Morianou I, Crisanti A, Nolan T, Hammond AM. CRISPR-Mediated Cassette Exchange (CriMCE): A Method to Introduce and Isolate Precise Marker-Less Edits. CRISPR J. 2022;5(6):868-76. Epub 2022/11/16. doi: 10.1089/crispr.2022.0026. PubMed PMID: 36378258.

4. Quinn C, Anthousi A, Wondji C, Nolan T. CRISPR-mediated knock-in of transgenes into the malaria vector Anopheles funestus. G3 (Bethesda). 2021;11(8). Epub 2021/12/02. doi: 10.1093/g3journal/jkab201. PubMed PMID: 34849822; PubMed Central PMCID: PMCPMC8496255.

Reviewer #4: Uploaded as an attachment.

Reviewer #5: The article describes proof of principle for one type of 'self-extinguishing' gene drive. The self-extinguishing nature is designed to come from internal cleavage of the drive element that can repair in a way that both removes the element and leaves in its place an (almost) wild type allele that is resistant to future gene drive activity. In reality the self-extinguishing nature of the element described and validated in this article is afforded by the negative fitness costs associated with the gene drive element itself, as well as its propensity to generate drive-resistant alleles.

I think there are elements of interest here for the gene drive field, in terms of the proof of principle of this type of self extinguishing element. However, I find this article to be a real mixed bag - elements of extraordinary clarity and lucidity interspersed with long sections of non-selective descriptions of results that are often incidental to the real points of interest and novelty. For example, the Introduction is one of the best I've read in a primary research article on gene drive. The Discussion, for the most part is not a Discussion but a (good) recapitulation of the results. The same cannot be said for the Results though - in parts these are impossible to follow and I say that as someone familiar with gene drive. They are over-long, non-selective and lack explainers. A crucial element that is missing, very early on in the article, is an overview of the approach accompanied by a schematic. I was two-thirds through the article before I realised the goal was to release a goldilocks element - just the right amount of power to increase in frequency for a bit but then ultimately falls away after its magic is done, thanks to this auto-excision and resistance. (I had initially thought the idea was to show that SSA can happen in a test strain, then release an element that could 'chase' a previous gene drive that had been stacked with pre-loaded direct repeats as a potential recall).

I use the term Goldilocks deliberately because it is precious and prevaricated on quite a few things being 'just right' and so does not escape the uncertainty (just as other gene drives do not) in predicting how this will actually behave. I think this element deserves more expansion in the Discussion.

Rather than list all my comments with regards to structure and clarity here I have included an annotated version - it deliberately has some of my errors of understanding left in as I think these will usefully highlight where some problems can arise and I hope the authors will find it useful.

**Have all data underlying the figures and results presented in the manuscript been provided?**

Reviewer #1: Yes

Reviewer #2: Yes

Reviewer #3: None

Reviewer #4: Yes

Reviewer #5: Yes

PLOS authors have the option to publish the peer review history of their article (what does this mean?). If published, this will include your full peer review and any attached files.

Reviewer #1: No

Reviewer #2: No

Reviewer #3: No

Reviewer #4: No

Reviewer #5: No

---

## [Decision Letter · Decision Letter 1]

8 Jul 2024

Dear Dr Myles,

Thank you very much for submitting your Research Article entitled 'Repeat mediated excision of gene drive elements for restoring wild-type populations' to PLOS Genetics.

The manuscript was fully evaluated at the editorial level and by independent peer reviewers. The reviewers appreciated the attention to an important topic but identified some concerns that we ask you address in a revised manuscript.

We therefore ask you to modify the manuscript according to the review recommendations. Your revisions should address the specific points made by each reviewer.

Yours sincerely,

Jackson Champer

Academic Editor

PLOS Genetics

Gregory Copenhaver

Section Editor

PLOS Genetics

The authors have fixed most of the issues, and the manuscript is not likely to go out for review again. However, some issues remain from reviewer 3, which are hopefully clearer in the new review. This reviewer presents some options that the authors could use to really thoroughly investigate their constructs. They should seriously consider these, but perhaps the authors could use their existing constructs to address at least some of this, which will likely be enough for the publication.

This could help determine if the wild-type alleles (which should be labeled more clearly as such in the figures, and perhaps be added as an outcome to Figure 1C because they are much more common) actually come from the ReMEDE element or if the ReMEDE just has lower cut rates compare to the original yMCR. I’d lean toward the latter as the more likely explanation, with the gypsy elements perhaps reducing Cas9 expression, or the ReMEDE removing the drive before it has the chance to express enough Cas9 to cut all the wild-type alleles. These possibilities will need to be clearly laid out if there is not a clear experimental answer.

As noted in reviewer 3’s point #1, the experiment would involve using males, ideally separate experiments with both constructs. Based on the promoters, it looks like everything would express quite well in males (both vasa and nanos have high male expression, nanos might even be higher in males than females in many Cas9 constructs). Thus, you could just cross ReMEDE males to wild-type or yellow (from resistance alleles) females and check the outcomes. With no drive-induced cleavage or wild-type template to track, you could be more certain of SSA outcomes, which could address some of what reviewer 3 is concerned about. Using ReMEDE homozygous females could also work well for this purpose, especially if there was a problem with this system in males for unclear reasons.

For the comments of reviewer 2 and the other comments of reviewer 3, changes to the text for clarity and emphasis should be sufficient. For aspects of reviewer 3’s comments on experiments and uncertainty, it will be important to address it as much as possible, probably including the above suggested experiment.

Finally, (for authors and reviewers) in the manuscript pdf, the figure resolutions were indeed very low. However, this was because the authors submitted separate figure files that were then converted by the online system to be part of the pdf, reducing the resolution in the process. To rectify this, you need to click on the “tif” file download in the upper right corner of the figure pages. These will then download the higher resolution figures, which are legible. However, I should note that while they are legible, it may still not meet the journal standards, so consider uploading higher resolution versions as part of the revisions, or else you might be asked to do it again anyway later in the publication process.

Sincerely,

Jackson Champer

Reviewer's Responses to Questions

**Comments to the Authors:**

Reviewer #1: The authors have addressed all my comments comprehensively, and the manuscript has been improved in current version with necessary revisions and extra sequencing data, which enhanced the accuracy and clarity of their work. Therefore, I recommend to accept this manuscript for publication in PLOS Genetics.

Reviewer #2: Thank you for your responses. The idea of using SSA to construct a self-extinguishing gene drive is interesting and this paper serves as a good proof of principle. I only have one minor comments now: regarding the sequencing result in Fig.3A, it's good to see evidence of resistance forming over time but only at later stage, however I found it hard to understand the actual frequency of this 17bp deletion. The caption said it's the representative sequences and on figure 3A the 'samples' stat for this sequence is 3. Does that mean you found three individuals bearing the same 17bp deletion? If yes, how many samples did you sequenced to find these 3? Would be good to clarify this.

Reviewer #3: Review uploaded as an attachment

Reviewer #4: The authors have revised the manuscript sufficiently. I appreciate that the authors took the time to carefully address my all comments. This revised version can be accepted for publication. however, the authors should provide a better quality images in the final version of the manuscript. Images/pictures in the present version are not of great quality.

Reviewer #5: The authors had an extensive set of reviews, a common thrust of which was to improve the clarity of the exposition of the results, to do the work justice. Looking at the responses to reviewers I am happy with the clarifications. I must admit, I have not had the time to re-read the whole article and compare the before and after, so I am taking the authors on their word that they have incorporated the necessary changes.

The low resolution of the figures in the resubmitted PDF precluded me from looking at them.

**Have all data underlying the figures and results presented in the manuscript been provided?**

Reviewer #1: Yes

Reviewer #2: Yes

Reviewer #3: None

Reviewer #4: Yes

Reviewer #5: Yes

PLOS authors have the option to publish the peer review history of their article (what does this mean?). If published, this will include your full peer review and any attached files.

Reviewer #1: No

Reviewer #2: No

Reviewer #3: No

Reviewer #4: No

Reviewer #5: No

---

## [Editor Report · Decision Letter 2]

4 Oct 2024

Dear Dr Myles,

We are pleased to inform you that your manuscript entitled "Repeat mediated excision of gene drive elements for restoring wild-type populations" has been editorially accepted for publication in PLOS Genetics. Congratulations!

Yours sincerely,

Jackson Champer

Academic Editor

PLOS Genetics

Gregory P. Copenhaver

Section Editor

PLOS Genetics

Comments from the reviewers (if applicable):

It looks like SSA is nicely confirmed, though with a little more trouble than anticipated. Still, I certainly think that the work is publishable now, and I’m happy to accept it. Kudos to the authors for getting through some extensive reviews, and special thanks to the reviewers, who were very helpful for this manuscript.

**Data Deposition**

http://datadryad.org/submit?journalID=pgenetics&manu=PGENETICS-D-23-01355R2

**Press Queries**

---

## [Editor Report · Acceptance letter]

4 Nov 2024

PGENETICS-D-23-01355R2 

Repeat mediated excision of gene drive elements for restoring wild-type populations 

Dear Dr Myles, 

We are pleased to inform you that your manuscript entitled "Repeat mediated excision of gene drive elements for restoring wild-type populations" has been formally accepted for publication in PLOS Genetics! Your manuscript is now with our production department and you will be notified of the publication date in due course.

With kind regards,

Anita Estes

PLOS Genetics

On behalf of:
